# How Large language models implement chain-of-thought?

## Abstract

Chain-of-thought (CoT) prompting has showcased the significant enhancement in the reasoning capabilities of large language models (LLMs). Unfortunately, the underlying mechanism behind how CoT prompting works remains elusive. Advanced works show the possibility of revealing the reasoning mechanism of LLMs by leveraging counterfactual examples (CEs) to do a causal intervention. Specifically, analyzing the difference between effects caused by reference examples (REs) and CEs can identify the key attention heads related to the ongoing task, *e.g.*, a reasoning task. However, the completion of reasoning tasks involves diverse abilities of language models such as numerical computation, knowledge retrieval, and logical reasoning, posing challenges to constructing proper CEs. In this work, we propose an in-context learning approach to construct the pair of REs and CEs, where REs can activate the reasoning behavior and CEs are similar to REs but without activating the reasoning behavior. To accurately locate the key heads, we further propose a word-of-interest (WoI) normalization approach to filter out irrelevant words and focus on the ones related to a specific reasoning task. Our empirical observations show that only a small fraction of attention heads contribute to the reasoning task. Performing interventions on these identified heads can significantly hamper the model's performance on reasoning tasks. Among these heads, we find that some play a crucial role in *judging* for a final answer, and some play a crucial role in *synthesizing* the step-by-step thoughts to get answers, which corresponds to the two stages of the CoT process: firstly think step-by-step to get intermediate thoughts, then answer the question based on these thoughts.

## 1 Introduction

Significant advancements have been observed in the field of large language models (LLMs)(OpenAI, 2023; Zhao et al., 2023). By formulating tasks in natural language instructions, LLMs have exhibited the capability to undertake single-step natural language processing (NLP) tasks. Yet LLMs even at the scale of 100 are struggling with handling multi-step reasoning tasks (Wei et al., 2022). Recent research works (Wei et al., 2022; Kojima et al., 2022; Zhang et al., 2022) focused on Chain-of-Thought (CoT) significantly improve the models' performance on reasoning tasks. By adjusting the input text prompt, the CoT technique elicits the model to generate multiple reasoning steps (chain-of-thought) preceding the final answer. As LLMs are increasingly being deployed to real-world scenarios, there is a general agreement on the importance of endowing LLMs with explainability, aiming at verifying and understanding the internal reasoning capabilities of LLMs.

Considerable efforts (Hanna et al., 2023; Wang et al., 2023) have been directed towards elucidating the underlying mechanism of specific tasks in small-scale LLMs such as GPT-2 small. These investigations focus on understanding the computational process of GPT-2 small for tasks such as "greater than" calculations (Hanna et al., 2023) and indirect object identification (Wang et al., 2023) using a method called "path patching". Initially introduced in (Wang et al., 2023), path patching is a methodology that identifies the key components and circuits within the model to facilitate specific tasks. The path patching method employs the construction of counterfactual examples (CEs) through the lens of causal intervention, drawing inspiration from the field of causal inference (Pearl, 2009). CEs represent hypothetical scenarios that deviate from reality in certain aspects, enabling

the inference of causal relationships by comparing the outcomes of reference and counterfactual scenarios.

While the interpretability theory has become well-established, the literature lacks an exploration of the interpretability of the LLM's CoT reasoning capabilities, which results from three challenges. First, the search space for identifying crucial modules significantly expands with the increase in model parameters, posing substantial challenges to the search process. Second, the mechanism behind the model's reasoning capabilities is highly intricate, as it requires the utilization of various abilities such as numerical computation, knowledge retrieval, and logical reasoning. Therefore the critical modules and circuits to perform reasoning tasks can be highly complex. Third, identifying the key modules responsible for CoT reasoning depends on comparing the outcomes of REs and CEs, while LLMs have an extremely large number of potential words. This leads to a sparse causal effect that is hard to detect since the words related to CoT reasoning are extremely few.

To address these challenges, we propose a novel framework for large-scale LLMs by constructing appropriate counterfactual examples paired with an interest-focused approach. Specifically, we leverage two state-of-the-art large language models, i.e., LLaMA2-7B (Touvron et al., 2023) and Qwen-7B (Bai et al., 2023)), to investigate the CoT reasoning mechanism. The large size ensures strong reasoning ability. In addition, we employ in-context learning (Brown et al., 2020) to construct reference and paired counterfactual examples. To tackle the sparse cause-effect issue, we propose a word-of-interest (WoI) normalization method, which filters out irrelevant words and focuses solely on those relevant to the predictions of a specific reasoning task. This approach facilitates the identification of the key modules responsible for CoT reasoning.

We conduct experiments using the publicly available Qwen-7B (Bai et al., 2023) and LLaMA2-7B (Touvron et al., 2023) language models. We explored the key attention heads utilized in completing the CoT reasoning tasks across the three datasets. We find that only a small subset of the heads significantly impact the model's CoT reasoning capability. Furthermore, we observe that the important heads for all reasoning tasks are predominantly located in the middle and top layers of the model. To validate the importance (faithfulness) of the identified heads, we performed knockout experiments by knocking out these heads. As a result, the model's ability to accurately predict reasoning answers is significantly weakened.

Our main findings are: (1) Adjusting the few-shot examples within the CoT prompt is an effective approach for generating REs and CEs in path patching experiments. This technique enables the language model to generate multi-step reasoning contents either or refrain from doing so in a controlled manner; (2) We introduce an innovative method that facilitates identifying the key modules of LLMs: designing word-of-interest normalization to locate key modules specific to CoT reasoning;(3) Extensive experiments are conducted on diverse reasoning tasks, uncovering intriguing insights into the performance of LLMs in reasoning tasks.

## 2 BACKGROUND

**Chain-of-thought (CoT) prompting.** CoT prompting, initially introduced by (Wei et al., 2022), is a technique that enhances the reasoning capabilities of large language models. By incorporating rationales into examples, CoT prompting prompts the models to generate reasoning content before providing an answer, thereby enhancing LLMs' reasoning ability. Kojima et al. (2022) introduce zero-shot CoT prompting, which enables the model to output reasoning content simply by appending the phrase "Let's think step by step" to the prompt, eliminating the designing of task-specific examples and reasoning instances. Subsequent studies further improve the accuracy of LLMs on reasoning tasks by modifying and selecting the generated reasoning content, such as self-consistent CoT (Wang et al., 2022) and Tree-of-thoughts Yao et al. (2023). Through extensive experiments, we find that few-shot CoT is an effective way to control LLMs' reasoning ability and therefore we adopt the few-shot prompt approach to construct REs and CEs.

**Interpretability of Large Language models.** Despite the impressive capabilities of large language models, their internal mechanisms still lack comprehensive understanding. Within (Wang et al., 2023), the authors introduced the novel concept of path patching and applied this methodology to identify pivotal circuits within the GPT-2 small model (0.1 billion parameters) for indirect object identification (IOI) tasks. Subsequently, (Goldowsky-Dill et al., 2023) refined this technique in a

subsequent publication. Path patching emerges as a potent approach for scrutinizing critical components at various levels of granularity, encompassing attention heads and linear transformations within MLP layers, inherent to a given model. This methodology draws inspiration from causal mediation analysis, entailing perturbations in component inputs and subsequent observation of the consequential changes in the model's behavior. In Hanna et al. (2023), path patching is implemented to delve into the intricacies of GPT-2 small's handling of greater-than tasks. Wu et al. (2023) takes the first step toward understanding the working mechanism of 7B-sized large language model. Method Distributed Alignment Search (DAS) Geiger et al. (2023) based on causal abstraction is applied to align the language model with a hypothesized causal model. Nevertheless, a dearth of LLM interpretability research currently exists. The key components crucial for undertaking more intricate tasks, like reasoning, remain largely unexplored. Due to the complexity of CoT reasoning tasks, it's intricate to design a unified symbolic causal model for multi-step reasoning tasks. In this work, employing the path patching method from (Wang et al., 2023; Hanna et al., 2023), we successfully identify the crucial attention heads in LLMs responsible for CoT reasoning ability. To validate the identified attention heads, one approach is to compare the overall behavior of the complete model with that of a model identical in all aspects except for the absence of that specific head, which is referred to as "knockout" experiment. Previous work (Wang et al., 2023) has indicated that replacing the activation of the attention head with an average activation value across some reference distribution is an effective knockout approach.

**Large language models (LLMs).** Qwen-7B (Bai et al., 2023) and LLaMA2-7B (Touvron et al., 2023) are two pre-trained language models of 7 billion parameters. The model weights for both architectures are openly accessible and can be acquired from HuggingFace (Wolf et al., 2020). Training data for both models is composed of extensive English language corpora. Specifically, Qwen-7B is trained on a dataset encompassing 3 trillion tokens mainly from English and Chinese corpora, whereas LLaMA2-7B is predominantly trained on a dataset containing 2 trillion tokens from English corpora (comprising 89.7% of training data). In performance evaluation, both models exhibit remarkable proficiency in reasoning NLP tasks. In terms of model architecture, the two models closely resemble each other, adopting a decoder-only transformer with 32 layers and 32 attention heads per attention layer.

## 3 METHOD

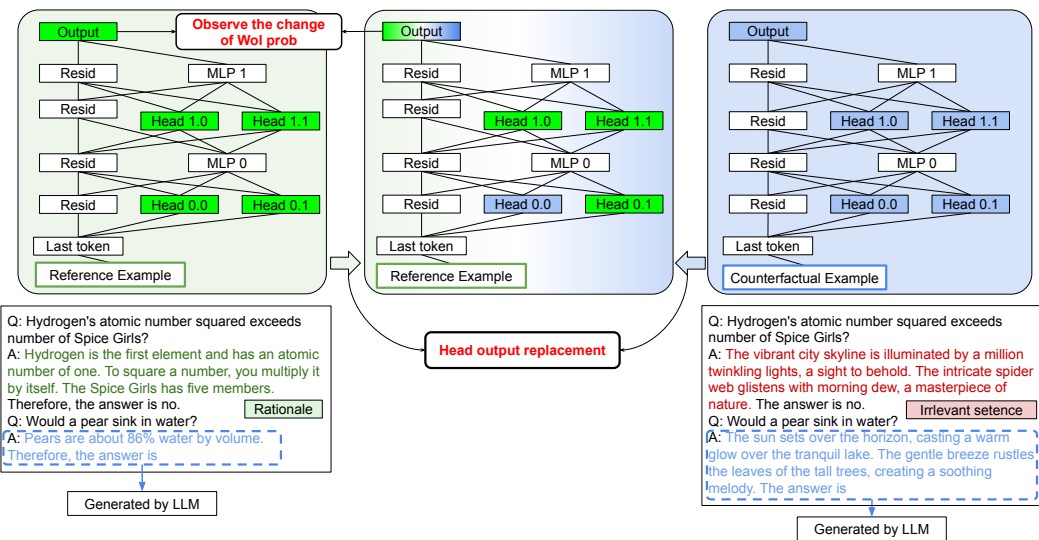

Figure 1: Overview of our method for identifying the key heads for LLMs completing CoT reasoning tasks.

In this section, we will detail the framework for identifying attention heads of LLMs that are crucial to CoT reasoning through path patching (Wang et al., 2023). An overview of our method is depicted in Figure 1, and our method mainly consists of the following parts: i) Sec. 3.1 identifies the key

attention heads using the path patching method; ii) Construct the REs and CEs for path patching in Sec. 3.2; iii): Evaluate the causal effect using the proposed word-of-interest normalization in Sec. 3.3; and iv) Verify the key attention heads through a knockout operation in Sec. 3.4.

## 3.1 IDENTIFY KEY HEADS THROUGH PATH PATCHING

Path patching is a method for localizing the key components of LLMs for accomplishing specific tasks or behaviors. The term "behavior" is defined within the context of input-output pairs on a specific dataset, thereby effectively reflecting the model's specific capabilities. To assess the significance of a model component for a particular behavior, a causal intervention is implemented to interfere with the behavior. Assuming that the component holds importance, the causal effect on the model's specific ability should be notable. The causal intervention applied to the model component is achieved through the creation of appropriate reference examples (REs) and counterfactual examples (CEs), which solely vary in their capacity to trigger the model's specific behavior.

Specifically, to apply causal intervention to a specific attention head, given a pair of inputs $x_r$ and $x_c$ (we use $x_r$ and $x_c$ to denote REs and CEs respectively), a forward pass is run on $x_r$, but replace the output of a specific attention head $i.j$ (head $i.j$ represents the $j$-th attention head in $i$-th layer of the model) with those from $x_c$. The layers after $i$ are recomputed as in a normal forward pass while keeping the output of the other heads unchanged. As shown in fig 1, for illustration, we simplify the architecture of the language model to a two-layer transformer. The importance of Head $0.0$ can be determined by checking the causal effect caused by the intervention in model output [1].

In this study, we aim to localize the key attention head within the model responsible for CoT reasoning. However, we are faced with two primary challenges: i) The complexity arises from the diverse range of abilities that LLMs possess to achieve CoT reasoning, including numerical computation, knowledge retrieval, and logical reasoning, making it challenging to design reference examples (REs) that are accurately paired with counterfactual examples (CEs) differing only in their ability to trigger the CoT reasoning behavior of LLMs. ii) Additionally, the extensive potential word in the LLM's output results in a sparsity of causal effects, as the words directly related to CoT reasoning are significantly limited in number. To overcome these challenges, we propose an innovative approach for constructing paired REs and CEs, accompanied by a word-of-interest (WoI) normalization method aimed at addressing the issue of sparse causal effects.

## 3.2 CONSTRUCT REs AND CEs FOR PATH PATCHING

To construct reference examples $x_r$ and counterfactual examples $x_c$ for path patching, we set $x_r$ to induce the model's normal CoT reasoning behavior, while $x_c$ is used to perturb the model's reasoning performance while attempting to maintain other information unchanged as much as possible. $x_r$ and $x_c$ are generated through the following steps: Firstly, we sample a question from a CoT dataset. Next, we generate $x_r$ by applying a specifically designed few-shot prompt template to this question. Finally, we generate $x_c$ by modifying some parts of $x_r$, noting that one $x_r$ can produce multiple different $x_c$. Once the generation process is complete, we perform data filtering to retain the data where the model provides a correct answer on $x_r$ but produces an incorrect answer on the corresponding $x_c$.

**CoT task and datasets.** We sample questions and conduct our experiment on 3 benchmark datasets for evaluating the effectiveness of CoT: **(1)** the **StrategyQA** (Geva et al., 2021) dataset which necessitates models to infer a multi-hop strategy for answering common sense questions, obtained from the BIG-bench (Srivastava et al., 2022), **(2)** the **AQuA** (Ling et al., 2017) dataset, which consists of algebraic and math reasoning problems, and **(3)** the **CSQA** (Talmor et al., 2018) dataset, which involves answering commonsense questions about the world with intricate semantics that often require prior knowledge and multi-step reasoning.

---

[1]We only manipulate the output of attention heads on the last token position of REs.

Table 1: Reference example for dataset CSQA and corresponding LLaMA2-7B output.

---

**REFERENCE EXAMPLE (MODEL INPUT):**

Q: What home entertainment equipment requires cable? Answer Choices: (a) radio shack (b) substation (c) cabinet (d) television (e) desk
A: The answer must require cable. Of the above choices, only television requires cable. Therefore, the answer is (d).
Q: To locate a choker not located in a jewelry box or boutique where would you go? Answer Choices: (a) jewelry store (b) neck (c) jewlery box (d) jewelry box (e) boutique
A:

**MODEL OUTPUT:**

The answer must be a place where you would go to locate a choker. Of the above choices, only a jewelry store is a place where you would go to locate a choker. Therefore, the answer is (a).

---

**Reference examples generation.** We generate $x_r$ by taking instances from the aforementioned CoT task datasets and applying a specially designed CoT prompt template on them to explicitly induce the reasoning behavior of the model. In this work, each reference data instance comprises two examples[2] following the ⟨`question, rationale, answer`⟩ format. Specifically, for the StrategyQA dataset, the provided reasoning process from the dataset itself is incorporated as the `rationale` (see Table 1). While for AQuA and CSQA datasets, we adopt the rationales written by (Wei et al., 2022). For each dataset, the same few-shot prompt is utilized, with only the final question being replaced. Table 1 shows a case of REs on CSQA and the output when feeding to LLaMA2-7B. See Appendix Table 3 for REs on dataset StrategyQA and AQuA.

Table 2: Counterfactual example for CSQA and corresponding LLaMA2-7B output.

---

**COUNTERFACTUAL EXAMPLE (MODEL INPUT):**

Q: What home entertainment equipment requires cable? Answer Choices: (a) radio shack (b) substation (c) cabinet (d) television (e) desk
A: The busy bee diligently collects nectar from one flower to another. The answer is (d).
Q: To locate a choker not located in a jewelry box or boutique where would you go? Answer Choices: (a) jewelry store (b) neck (c) jewlery box (d) jewelry box (e) boutique
A:

**MODEL OUTPUT:**

The playful dolphins leap and frolic in the sparkling blue ocean. The cat sleeps peacefully on the cozy cushion. The answer is (a).

---

**Counterfactual examples generation.** $x_c$ is generated by partially editing $x_r$ with the purpose of altering the model's CoT reasoning ability. In order to suppress the reasoning capability of the model while maintaining other information unchanged, we replace the `rationale` in $x_r$ with irrelevant sentences describing natural scenery, which were randomly generated using ChatGPT-3.5 (Table 2). We ensure that the CEs and REs have the same token-level length. By adjusting `rationale` in the prompt, when faced with questions demanding logical inference, the models tend to produce irrelevant content instead of a chain-of-thought context[3] (Table 2). Table 1 shows a case of CEs on CSQA and the output when feeding to LLaMA2-7B. See Appendix Table 4 for CEs on dataset StrategyQA and AQuA. For both REs and CEs, we concatenate the model input and output (excluding the final answer token) to form a complete RE/CE.

---

[2]Due to space limitations, we only show one QA pair in the prompt for the RE and CE case displayed in Table 1 and 2. In the actual experiment, we use two QA pairs. Please refer to Section B for the complete example

[3]We ensure that the generated irrelevant sentences are simple declarative statements and do not involve any inference or coherence relationships between them. Furthermore, the reasoning involved in the dataset we use is unrelated to natural landscapes, which mainly focuses on mathematical, commonsense, and historical reasoning. Therefore, the likelihood of the replaced content affecting the reasoning outcomes is minimal.

### 3.3 WORD-OF-INTEREST NORMALIZATION

The evaluation of each attention head's importance involves the introduction of causal intervention and subsequent analysis of the resulting alterations in the model's final layer output. However, the large number of potential words in the final output presents challenges in assessing the change in the model's CoT reasoning ability. Therefore, it is imperative to develop appropriate metrics for evaluating the variations in the model's output. Experimental investigations reveal that distinct metric design approaches result in the identification of different heads (see Section D). We propose a novel method called Word-of-Interest normalization to effectively evaluate the discrepancies caused by path patching. "Word-of-Interest" refers to the selective consideration of output logits exclusively associated with tokens related to answer options. "Normalization" involves the utilization of token probabilities obtained after softmax transformation. Subsequently, the evaluation of discrepancy involves the computation of the change of metric: $t = \frac{p_{gt}}{\sum p_{cd}}$, where $p_{gt}$ indicates the probability of the token corresponding to the ground truth answer, $\sum p_{cd}$ denotes the sum of probabilities for all candidate tokens (e.g., in the CSQA dataset, $\sum p_{cd}$ means the sum of probability of all option tokens "*a, b, c, d, e*"). This metric quantifies the likelihood of the model selecting the correct answer from a set of all possible options. Based on the designed metric, the rate of change of metric $t$ serves as the evaluative metric to assess the importance of the attention head $i.j$: $\alpha^{h_{i,j}} = \frac{t_c^{h_{i,j}} - t_o}{t_o}$, where $t_o$ denotes the $t$ before path patching and $t_c^{h_{i,j}}$ denotes the $t$ after path patching on head $i, j$. The negative value of $\alpha^{h_{i,j}}$ means a drop of relative confidence in the ground truth answer compared to other candidates, thus indicating impaired reasoning performance and a larger drop corresponds to the higher importance of head $i, j$. For each pair of REs and CEs, we run path patching for all heads of each layer in the model and sort heads by the value of $\alpha^{h_{i,j}}$ to identify the top $k$ key heads.

### 3.4 VERIFY KEY HEADS THROUGH KNOCKOUT EXPERIMENTS

After identifying the key attention heads using path patching, we verify these heads' impact on the model's CoT reasoning ability. Our methodology entails the "knockout" of the top $k$ most important heads obtained via path patching. Suppose the model's reasoning ability remains unaffected after randomly knocking out heads but demonstrates a significant deterioration when the top $k$ heads are knocked out. In that case, it is verified that the identified heads facilitate reasoning tasks. The overview of knockout method is shown in Fig 6.

In the knockout procedure, we substitute the outputs of attention heads at each token position in the RE with the respective outputs from the CE. Specifically, starting from the last token and moving backward, we replace the output of the model's top $k$ heads on $x_r$ with that on $x_c$, as depicted in Fig 6[4]. We then observe the variations in the model's predictions for the "[label]" before and after the replacement. We evaluate the change in the model's reasoning capability using the metric **prediction accuracy** $Acc$.

## 4 EXPERIMENTS

To interpret the CoT reasoning ability of the model into human-understandable components, we perform experiments as follows: (1) *discover* the key components in completing the CoT reasoning task by path patching in Section 4.1; (2) *validate* the faithfulness of the important components by knockout analysis in Section 4.2; (3) *understand* the behavior of the newly identified components by examining their attention patterns in Section 4.3; (4) Discuss the different methods of constructing REs and CEs in Section 4.4.

### 4.1 KEY HEAD DISTRIBUTION

Based on the REs/CEs constructed in Section 3.2 and metric designed in Section 3.3, we successfully identify the key attention heads in LLaMA2-7B and Qwen-7B for making correct predictions on CoT reasoning tasks. The distribution of these heads is depicted in Figure 2, where the magnitude of each point represents the change in the probability of the correct answer (*e.g.*metric $t$) after perturbing

---

[4]When token length of $x_c^*$ is shorter than that of $x_r^*$, we leave the first $l_{x_r^*} - l_{x_c^*}$ tokens not replaced, where $l_{x_r^*}, l_{x_c^*}$ are the length of augmented reference data and augmented counterfactual data respectively.

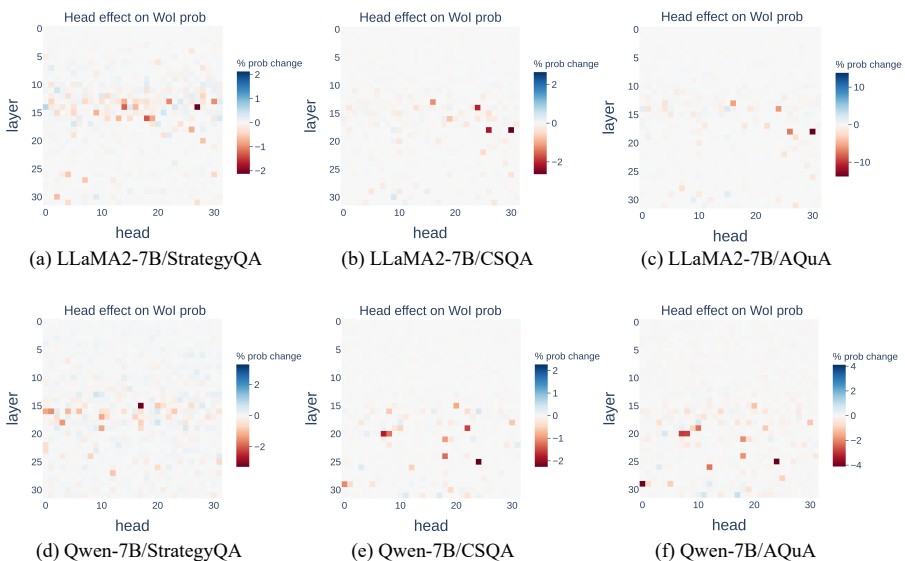

Figure 2: Identified key heads of LLaMA-7B and Qwen-7B on StrategyQA, CSQA, and AQuA.

the corresponding head. Red color indicates a decrease in $t$ after perturbation, with darker shades indicating greater importance of the head. Several interesting properties can be observed from the two models:

(i) The key heads for the StrategyQA task are distributed in the middle layers of both models. In LLaMA2-7B, the key attention heads for CoT reasoning tasks are also distributed in the middle layers, while in Qwen-7B, they are concentrated in the later layers. This phenomenon is consistent with the general intuition, that lower layers are responsible for token processing, while higher layers for more complicated reasoning tasks. (ii) The impact of the most key attention heads in the AQuA task is notably larger compared to the strategyQA and CSQA tasks. We attribute the discrepancy to the intricate nature of the AQuA task, which involves complex multi-step numerical reasoning. The model heavily depends on the reasoning process to generate the correct answer. Consequently, perturbing the attention heads that contain crucial reasoning information exerts a more substantial influence. (iii) Both models exhibit a common phenomenon that some key attention heads are shared for completing CSQA and AQuA tasks. From these shared heads, we find that head $18.30$ in LLaMA2-7B and head $21.18$ in Qwen-7B mainly attend to the token corresponding to the correct answer. A detailed analysis of the behavior and characteristics of the identified key heads is provided in Section 4.3 and Appendix C.

## 4.2 VALIDATION OF KEY HEADS

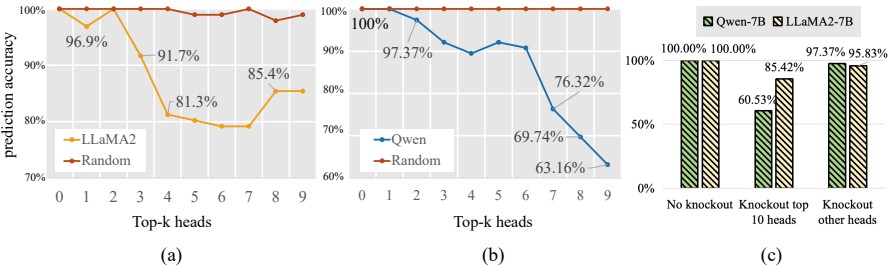

Figure 3: Validation of the key heads of LLaMA2-7B and Qwen-7B on CSQA dataset through knockout. (a) Change in prediction accuracy after knocking out the top-$k$ attention heads and randomly selected $k$ attention heads. (b) Change of model prediction accuracy when knocking out top 10 heads and other $1,014$ heads.

To validate the key attention heads we find, following the method proposed in section 3.4, we sequentially incorporate the top 9 important heads into the set of knocked-out heads in order of decreasing importance and observe the corresponding changes in the model's capabilities. The knockout results of LLaMA2-7B and Qwen-7B on the CSQA dataset are shown in Figure 3. The experiment involved 96 instances of CSQA $x_r$ data, where the model correctly predicted the answers. The results clearly indicate that knocking out the top 10 attention heads leads to a large drop in the model's accuracy (14.58%) in completing the CSQA reasoning task. While randomly perturbing nine attention heads, the model's predictive capability remains largely unaffected (accuracy only decreased by less than 5%).

### 4.3 Attention pattern of the Key Heads

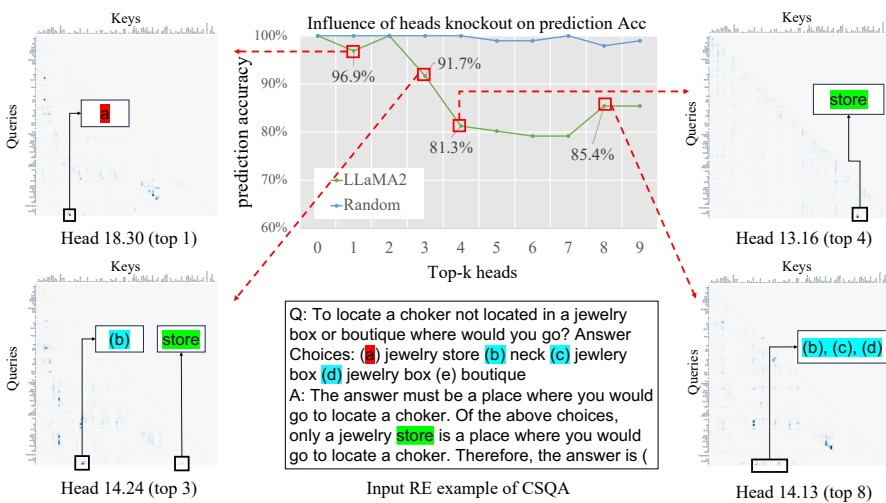

Figure 4: Attention map of head 18.30/14.24/13.16, and 14.13 in LLaMA2-7B on CSQA.

As depicted in Fig. 2, attention head 18.30 within the LLaMA2-7B model plays a pivotal role in processing both CSQA and AQuA datasets. Consequently, we postulate that this specific head directs its attention toward tokens that are associated with the options (such as $a, b, c, d$). Through visualizing the attention weights of head 18.30 towards the token immediately preceding the generated answer, we uncover a notable pattern of strong attention towards the token corresponding to the correct option. As illustrated in Fig. 4 (head "0" in Fig. 3 (a)), the model attends strongly to the answer token "(a)" with an attention weight 0.82, which means this top-1 head plays a key role in *judging* the final correct answer.

Similarly, we check the attention weights of head 13.16 on input tokens in LLaMA2-7B, which causes a large performance drop of the model after knockout (Fig. 3). This head strongly attends to the token "store" in the generated step-by-step thoughts. As "store" is also presented in the final correct answer "(a) jewelry store" and is the only different token compared with other options, this head plays a key role in synthesizing thoughts to get answers. *Analogically, the function of Head 18.30 and 13.16 corresponds to the two stages of the chain-of-thought (CoT) process: firstly think step-by-step to get intermediate thoughts, then answer the question based on these thoughts.*

Moreover, we also check the attention of head 14.13 on input tokens (head "8" in Fig. 3 (a)), which causes a surprising performance up after knockout. This heads strongly attends to other candidates and wrong options "(b),(c),(d)", the knockout of these heads diminishes the noisy factor from other options, so it's reasonable to get performance up by blocking it. We also observe a similar phenomenon on Qwen-7B (see Appendix C).

### 4.4 How to design REs/CEs for CoT reasoning?

In addition to the CE template mentioned in Section 3.2, we also experimented with some other CE templates:

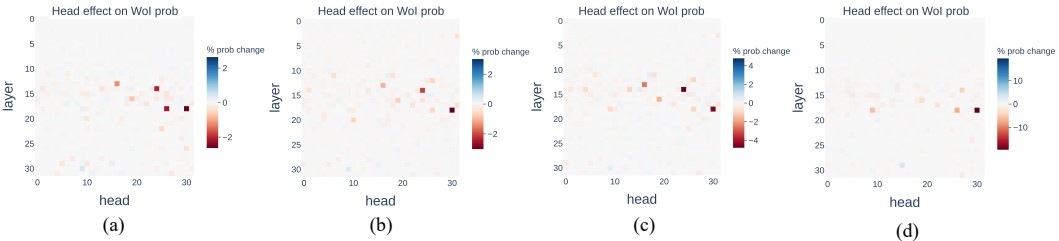

Figure 5: Key attention heads distribution using different REs/CEs: (a), (b), and (c): key attention heads distribution of LLaMA2-7B on CSQA using proposed CE templates (Section 3.2), template 1 and template 2, respectively. (d) key attention heads distribution of LLaMA2-7B on StrategyQA by transforming proposed StrategyQA REs and CEs into multi-choice format (template 3).

Template 1: Move the irrelevant statement to the beginning of the input content (see Table 5 case of template 1). In that case, the model directly outputs the answer instead of producing irrelevant statements first.

Template 2: Use a standard few-shot prompt by directly removing the `rationale` from the ⟨`question`, `rationale`, `answer`⟩ triplet of the few-shot prompts (see Table 5 case of template 2). Unrelated additional information can be eliminated by using this approach.

Template 3: For StrategyQA, transform the original StrategyQA question-answering (Table 3) input into a multiple-choice format (Table 6 case of template 3). Given our findings of attention heads that selectively attend to the correct options in both the CSQA and AQuA datasets, and considering the absence of option information in the original StrategyQA REs and CEs (see Table 3), we aim to explore whether these same crucial attention heads can be identified when the data is transformed into a multiple-choice format.

It is notable that the distributions of attention heads obtained from the three templates (Fig. 5a, b, and c) are similar. We observe that the amplitude of the key head in template 1 (Fig. 5b) slightly surpasses that of the proposed template (Fig. 5a). We posit that this discrepancy arises from the inherent limitation imposed on the model, whereby it is restrained from directly generating explicit answers, consequently compromising its reasoning capabilities. Moreover, the standard few-shot template (Fig. 5c) yields a larger amplitude and this amplitude tends to increase as the number of few-shot examples increases. We attribute this phenomenon to the fact that increasing the number of few-shot examples can further suppress the model's reasoning ability. As shown in Fig. 5d, when incorporating option information into the CEs and REs of StrategyQA, the identified key head (Head 18.30) aligns with that of CSQA and AQuA. We also explore designing REs and CEs base on zero-shot CoT (Kojima et al., 2022), see Appendix E for more results.

In addition, we conducted extensive experimentation to devise a suitable metric for path patching. Appendix D contains the key attention head distributions identified by various metrics.

## 5    CONCLUSION

In this study, we have identified the crucial components responsible for the reasoning ability of two 7B LLMs. We conduct path patching experiments with carefully designed REs/CEs derived from three CoT datasets and WoI norm metrics. The results of our experiments strongly support the notion that these particular heads play a vital role in the CoT reasoning ability of LLMs. More discussion about future work is shown in Appendix F.

Moreover, through additional analysis, we discover interesting patterns from the two of the identified key heads specifically. Additionally, through extensive ablation studies on various CE templates and metrics to assess the importance of the patched heads, we find that in-context learning is an effective method to construct REs/CEs for path patching, which can control the behavior of LLMs. The control over LLMs can be strengthened by increasing the few-shot examples in the prompt. Focusing on the probability of the tokens related to a specific task/behavior is a useful method to deal with the sparse causal effect problem.

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

## A  APPENDIX: KNOCKOUT METHOD ILLUSTRATION

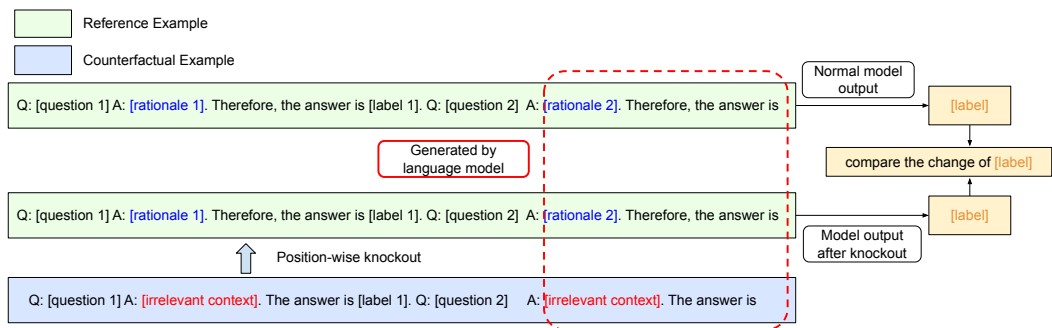

Figure 6: Knockout method for counterfactual data.

## B  APPENDIX: MORE REFERENCE/COUNTERFACTUAL EXAMPLES

Table 3: Reference examples for dataset CSQA/AQuA and the corresponding output of LLaMA2-7B

---

**REFERENCE EXAMPLE FOR STRATEGYQA (MODEL INPUT:)**

Q: Is it common to see frost during some college commencements?
A: College commencement ceremonies often happen during the months of December, May, and sometimes June. Frost isn't uncommon to see during the month of December, as it is the winter. Therefore, the answer is yes.
Q: Hydrogen's atomic number squared exceeds number of Spice Girls?
A: Hydrogen is the first element and has an atomic number of one. To square a number, you multiply it by itself. The Spice Girls has five members. Therefore, the answer is no.
Q: Could a llama birth twice during War in Vietnam (1945-46)?
A:

**MODEL OUTPUT:**

The War in Vietnam (1945-46) was a war between the Viet Minh and the French Union. The Viet Minh was a communist revolutionary group that fought for independence from France. The French Union was a group of French colonies that fought for independence from the Viet Minh. The answer is no.

**REFERENCE EXAMPLE FOR AQUA (MODEL INPUT:)**

Q: John found that the average of 15 numbers is 40. If 10 is added to each number then the mean of number is? Answer Choices: (a) 50 (b) 45 (c) 65 (d) 78 (e) 64
A: If 10 is added to each number, then the mean of the numbers also increases by 10. So the new mean would be 50. Therefore, the answer is (a).
Q: A person is traveling at 20 km/hr and reached his destiny in 2.5 hr then find the distance? Answer Choices: (a) 53 km (b) 55 km (c) 52 km (d) 60 km (e) 50 km
A: The distance that the person traveled would have been 20 km/hr * 2.5 hrs = 50 km. Therefore, the answer is (e).
Q: Of the 200 students in a school, at least 45% attended the prom night and at least 35% took part in the debating session. What is the maximum number of students who could have neither attended the prom night nor the debating session? Answer Choices: (a) 27 (b) 81 (c) 90 (d) 99 (e) 110
A:

**MODEL OUTPUT:**

The maximum number of students who could have neither attended the prom night nor the debating session would be 200 - (45% * 200) - (35% * 200) = 110. Therefore, the answer is (e).

---

Table 4: Counterfactual examples for dataset CSQA/AQuA and the corresponding output of LLaMA2-7B

---

**COUNTERFACTUAL EXAMPLE FOR STRATEGYQA (MODEL INPUT:)**

Q: Is it common to see frost during some college commencements?
A: The serene forest is alive with the sounds of chirping birds, their sweet melodies filling the air. The majestic eagle soars high above the mountains, surveying its domain. The answer is yes.
Q: Hydrogen's atomic number squared exceeds number of Spice Girls?
A: The vast desert stretches endlessly, its golden dunes shimmering in the sunlight. The vibrant rainbow paints the sky after a refreshing summer rain. The answer is no.
Q: Could a llama birth twice during War in Vietnam (1945-46)?
A:

**MODEL OUTPUT:**

The majestic lion roars as it stalks its prey, its powerful jaws clamping down on its unsuspecting victim. The answer is yes.

---

**COUNTERFACTUAL EXAMPLE FOR AQUA (MODEL INPUT:)**

Q: John found that the average of 15 numbers is 40. If 10 is added to each number then the mean of number is? Answer Choices: (a) 50 (b) 45 (c) 65 (d) 78 (e) 64
A: The tiny ant carries a heavy load several times its size. The icy glaciers glisten in the sunlight, showcasing their beauty and grandeur. The answer is (a).
Q: A person is traveling at 20 km/hr and reached his destiny in 2.5 hr then find the distance? Answer Choices: (a) 53 km (b) 55 km (c) 52 km (d) 60 km (e) 50 km
A: The tiny ant carries a heavy load several times its size. The icy glaciers glisten in the sunlight, showcasing their beauty and grandeur. The answer is (e).
Q: Of the 200 students in a school, at least 45% attended the prom night and at least 35% took part in the debating session. What is the maximum number of students who could have neither attended the prom night nor the debating session? Answer Choices: (a) 27 (b) 81 (c) 90 (d) 99 (e) 110
A:

**MODEL OUTPUT:**

The tiny ant carries a heavy load several times its size. The icy glaciers glisten in the sunlight, showcasing their beauty and grandeur. The answer is (a).

---

Table 5: CE cases of template 1 and 2 and the corresponding output of LLaMA2-7B

---

**CE OF TEMPLATE 1 FOR CSQA (MODEL INPUT:)**

The busy bee diligently collects nectar from one flower to another. The tranquil cabin by the lake is surrounded by towering pine trees, their scent filling the air with a soothing aroma.
Q: What do people use to absorb extra ink from a fountain pen? Answer Choices: (a) shirt pocket (b) calligrapher's hand (c) inkwell (d) desk drawer (e) blotter
A: The answer is (e).
Q: What home entertainment equipment requires cable? Answer Choices: (a) radio shack (b) substation (c) cabinet (d) television (e) desk
A: The answer is (d).
Q: To locate a choker not located in a jewelry box or boutique where would you go? Answer Choices: (a) jewelry store (b) neck (c) jewlery box (d) jewelry box (e) boutique
A:

**MODEL OUTPUT:**

The answer is (a).

---

**CE OF TEMPLATE 2 FOR CSQA (MODEL INPUT:)**

Q: What do people use to absorb extra ink from a fountain pen? Answer Choices: (a) shirt pocket (b) calligrapher's hand (c) inkwell (d) desk drawer (e) blotter
A: The answer is (e).
Q: What home entertainment equipment requires cable? Answer Choices: (a) radio shack (b) substation (c) cabinet (d) television (e) desk
A: The answer is (d).
Q: The sanctions against the school were a punishing blow, and they seemed to what the efforts the school had made to change? Answer Choices: (a) ignore (b) enforce (c) authoritarian (d) yell at (e) avoid
A: The answer is (a).
Q: To locate a choker not located in a jewelry box or boutique where would you go? Answer Choices: (a) jewelry store (b) neck (c) jewlery box (d) jewelry box (e) boutique
A:

**MODEL OUTPUT:**

The answer is (a).

---

Table 6: OE and CE cases of template 3 and the corresponding output of LLaMA2-7B

---

**OE OF TEMPLATE 3 FOR STRATEGYQA (MODEL INPUT):**

Q: Is it common to see frost during some college commencements? Answer Choices: (a) Yes (b) No
A: College commencement ceremonies often happen during the months of December, May, and sometimes June. Frost isn't uncommon to see during the month of December, as it is the winter. Therefore, the answer is (a).
Q: Hydrogen's atomic number squared exceeds number of Spice Girls? Answer Choices: (a) Yes (b) No
A: Hydrogen is the first element and has an atomic number of one. To square a number, you multiply it by itself. The Spice Girls has five members. Therefore, the answer is (b).
Q: Could a llama birth twice during War in Vietnam (1945-46)? Answer Choices: (a) Yes (b) No
A:

**MODEL OUTPUT:**

The War in Vietnam (1945-46) was a war between the Viet Minh and the French Union. The Viet Minh was a communist revolutionary group that fought for independence from France. The French Union was a group of French colonies that fought for independence from the Viet Minh. Therefore, the answer is (b).

---

**CE OF TEMPLATE 3 FOR STRATEGYQA (MODEL INPUT):**

Q: Is it common to see frost during some college commencements? Answer Choices: (a) Yes (b) No
A: The serene forest is alive with the sounds of chirping birds, their sweet melodies filling the air. The majestic eagle soars high above the mountains, surveying its domain. Therefore, the answer is (a).
Q: Hydrogen's atomic number squared exceeds number of Spice Girls? Answer Choices: (a) Yes (b) No
A: The vast desert stretches endlessly, its golden dunes shimmering in the sunlight. The vibrant rainbow paints the sky after a refreshing summer rain. Therefore, the answer is (b).
Q: Could a llama birth twice during War in Vietnam (1945-46)? Answer Choices: (a) Yes (b) No
A:

**MODEL OUTPUT:**

he majestic lion roars as it stalks its prey, its powerful jaws clamping down on its unsuspecting victim. Therefore, the answer is (a).

## C  APPENDIX: ATTENTION MAP ANALYSIS OF QWEN-7B ON CSQA

Figure 7: Attention map of head 19.22/20.8/21.18/15.20, and 18.30 in Qwen-7B on CSQA reference example.

For Qwen-7B, we conduct a similar analysis on the identified top 9 heads and discover five heads that exhibit distinct patterns, as shown in 7. Similar to the LLaMA2-7B, Qwen-7B also has a head (20.8) strongly attending to the correct answer options and a head (15.20) involved in synthesizing thoughts to obtain the answer. However, different from LLaMA2-7B, there are 2 attention heads of Qwen-7B attending to the correct answer options. Consistent with LLaMA2-7B, when the first head (20.8) attending to the correct answer options is knocked out, the model's predictive accuracy is not substantially affected. Nonetheless, in Qwen-7B, a remarkable decrease is observed when the second head dedicated to the correct answer options is knocked out. This discrepancy leads us to postulate the existence of additional backup heads in LLaMA2-7B, sharing a similar responsibility for attending to the correct answer options.

# D   APPENDIX: DIFFERENT METHODS OF DESIGNING METRIC

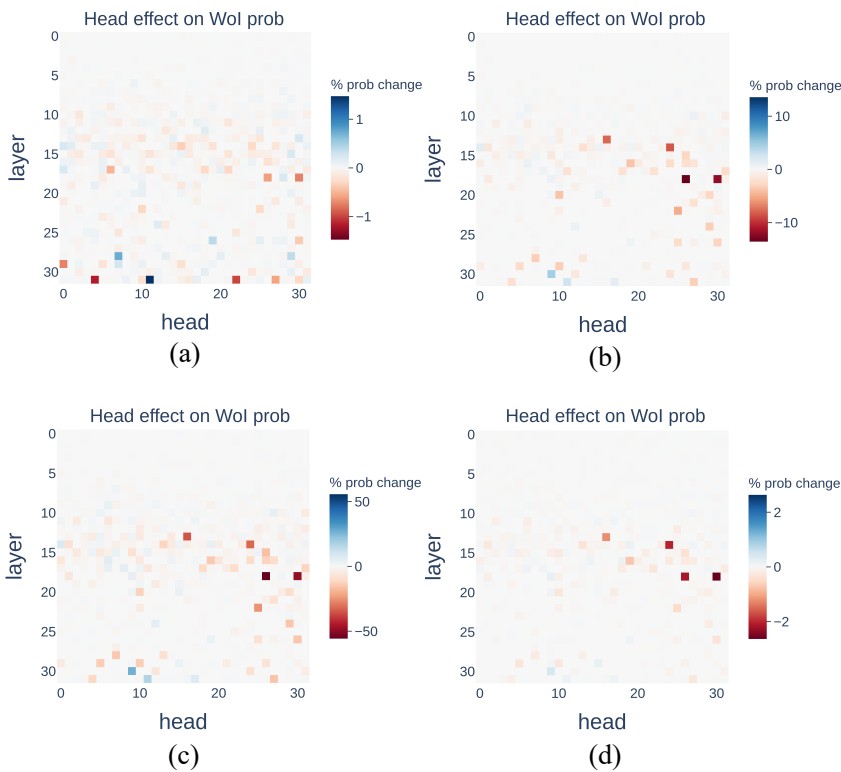

Figure 8: key attention heads distribution of LLaMA2-7B on StrategyQA using different metrics. (a) change rate of ground truth token logits. (b) Ground truth token logits minus the mean of logits values for other candidate tokens. (c) Divide ground truth token probability by the sum of probabilities of other candidate tokens. (d) Ground truth token probability divided by the sum of probabilities of all candidate tokens as proposed in Section 3.3.

Through empirical investigations, we find it is crucial to devise appropriate metrics for evaluating the causal effect caused by path patching. Improper metric formulation can impede the identification of pivotal attention heads. Furthermore, we discover that a sparse distribution of attention heads is a prerequisite for identifying key attention heads responsible for accomplishing CoT reasoning tasks.

During our preliminary endeavors, we explored various approaches to metric design. Intuitively, following the existing interpretability works, we adopt the change rate of logits on ground truth token as a metric, aiming to measure the influence on how the model outputs the accurate answer. However, we discover that the resulting distribution of heads using this metric is not sparse (as depicted in Figure 8(a)), and knocking out these heads did not substantially impact the model's reasoning capability. Therefore, we hypothesize that the token logits in the model's output contain a bias inherent to its question-answering capabilities. The attention heads identified using this metric may be those that influence this bias. To mitigate this bias, we devised three candidate metrics as shown in the caption of Fig. 8. The key attention heads located using these three metrics on LLaMA2-7B are shown in Figure 8 (b), (c), (d), respectively. It can be observed that the head distribution obtained from metric (d) is the best option for its sparsity of located key heads (ground truth token probability divided by the sum of probabilities of all candidate options). Therefore, we adopt this metric for our path patching experiment to locate the key attention heads.

# E    APPENDIX: SOME OTHER ATTEMPTS AT CONSTRUCTING REs/CEs

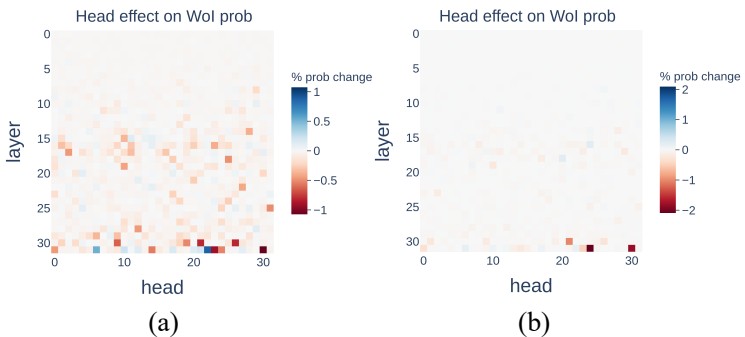

(a)                                                    (b)

Figure 9: Key attention heads distribution using different REs/CEs based on zeor-shot CoT. (a) Key attention heads distribution of Qwen-7B on StrategyQA using zero-shot CoT REs and CEs (adjusting the zero-shot CoT prompt). (b) Key attention heads distribution of Qwen-7B on StrategyQA using zero-shot CoT REs and CEs (adjusting the reasoning evidence).

In addition to few-shot CoT, we explore using zero-shot CoT (Kojima et al., 2022) method to activate the model's reasoning, which uses phrases like "let's think step by step" to prompt the model to generate reasoning content. We design CEs by interfere the zero-shot CoT prompt (e.g., replacing "let's think step by step" with "let's read word by word"). However, we find that the attention heads identified through this method are not sparse (see Figure 9 (a)), indicating an ineffective way to construct OEs/CEs for CoT reasoning.

Furthermore, on zero-shot CoT OEs/CEs, We also attempt to design CEs by interfering with the reasoning evidence of REs. In this case, the REs take the form of "[`question`]+[`reasoning evidence`]+Therefore, the answer is". We create CEs by randomly replacing keywords (such as proper nouns, verbs, adjectives, etc.) in the [`reasoning evidence`]. While this method allowed us to identify relatively sparse key heads (see Figure 9 (b)), the magnitude of changes in the heads was small (around $1\%$). Through knockout experiments, we observe that perturbing these heads does not significantly impact the model's reasoning abilities, which reveals that directly perturbing the reasoning evidence is not an effective way of identifying key heads for CoT reasoning.

## F APPENDIX: FUTURE WORK

**Transfer to other modalities**. We have used path patching to explore the CoT reasoning ability of LLMs. The reasoning ability of large models on other modalities (such as large language-vision models) remains under-explored. We hope our exploration of the textual modality can inspire research on the interpretability of reasoning abilities in other modalities.

**Analyze the MLP layers**. Due to the rich semantic information in attention heads, we identify and analyze the pattern of the key attention heads responsible for CoT reasoning. The role of MLP layers in CoT reasoning remains to be explored.

**Model editing.** Since we have located the attention heads responsible for CoT reasoning, it should be possible to modify the key attention heads through model training/editing to improve the model's reasoning capability.

**Transfer to generative tasks.** In this work, we set the REs/CEs to QA answering type. One promising direction is transferring our methodology to more generative reasoning tasks. However, it is challenging to i) design dynamic CEs to perform causal intervention and ii) evaluate the quality changes in generated content.

