## A    APPENDIX: PATH PATCHING AND KNOCKOUT METHOD

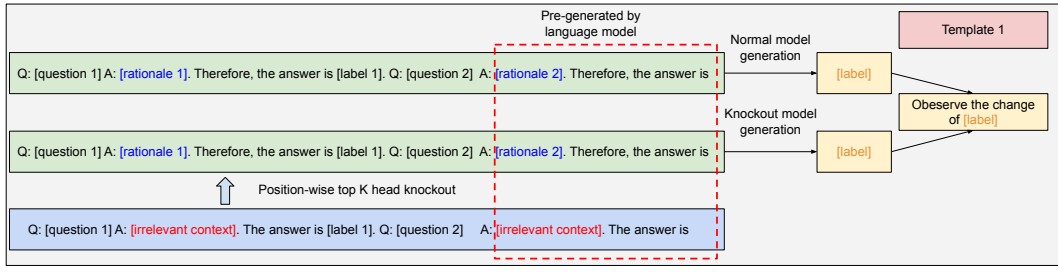

Figure 6: Overview of identifying the key heads for LLMs completing CoT reasoning tasks.

Figure 7: Knockout method for counterfactual data.

## B    APPENDIX: REFERENCE DATA EXAMPLES

## C    APPENDIX: INPUT/OUTPUT EXAMPLES OF COUNTERFACTUAL DATA

Table 4: Example of output results on counterfactual data produced by LLaMA-7B on StrategyQA.

**INPUT:**

Q: Is it common to see frost during some college commencements?
A: The towering redwood forest is a majestic sight, its ancient giants reaching towards the heavens. The moonlit beach shimmers in the distance, as the waves crash against the shore. The answer is yes.
Q: Hydrogen's atomic number squared exceeds number of Spice Girls?
A: The curious deer cautiously approaches a clear stream, its delicate nose sniffing the air for any potential danger. The tiny ant carries a heavy load several times its size. The answer is no.
Q: Would a rabbi worship martyrs Ranavalona I killed?
A:

**MODEL OUTPUT:** The majestic lion roars as it stares down its prey, its powerful jaws ready to strike. The tiny ant carries a heavy load several times its size. The answer is no.

Table 3: Reference data examples for dataset CSQA and AQuA

---

**REFERENCE DATA EXAMPLE FOR CSQA**

**Q:** What do people use to absorb extra ink from a fountain pen? Answer Choices: (a) shirt pocket (b) calligrapher's hand (c) inkwell (d) desk drawer (e) blotter

**A:** The answer must be an item that can absorb ink. Of the above choices, only blotters are used to absorb ink. Therefore, the answer is (e).

**Q:** What home entertainment equipment requires cable? Answer Choices: (a) radio shack (b) substation (c) cabinet (d) television (e) desk

**A:** The answer must require cable. Of the above choices, only television requires cable. Therefore, the answer is (d).

**Q:** Where is a bald eagle safe? Answer Choices: (a) pine tree (b) open country (c) in washington (d) wildlife refuge (e) sky

**A:**

---

**REFERENCE DATA EXAMPLE FOR AQUA**

**Q:** John found that the average of 15 numbers is 40. If 10 is added to each number then the mean of number is? Answer Choices: (a) 50 (b) 45 (c) 65 (d) 78 (e) 64

**A:** If 10 is added to each number, then the mean of the numbers also increases by 10. So the new mean would be 50. Therefore, the answer is (a).

**Q:** A person is traveling at 20 km/hr and reached his destiny in 2.5 hr then find the distance? Answer Choices: (a) 53 km (b) 55 km (c) 52 km (d) 60 km (e) 50 km

**A:** The distance that the person traveled would have been 20 km/hr * 2.5 hrs = 50 km. Therefore, the answer is (e).

**Q:** There are 10 oranges in a basket. Find the no. of ways in which 2 oranges are chosen from the basket? Answer Choices: (a) 45 (b) 90 (c) 120 (d) 150 (e) 180

**A:**

---

Table 5: Example of output results on counterfactual data produced by Qwen-7B on StrategyQA.

---

**INPUT:**

Q: Is it common to see frost during some college commencements?
A: The intricate spider web glistens with morning dew, a masterpiece of nature. The serene garden is a sanctuary of calm, its blooming flowers and chirping birds creating a peaceful atmosphere. The answer is yes.
Q: Hydrogen's atomic number squared exceeds number of Spice Girls?
A: The ancient ruins are a sight to behold, their weathered stones holding the secrets of the past. The old oak tree stands proudly in the middle of the meadow. The answer is no.
Q: Did Archduke Franz Ferdinand of Austria participate in the Pacific War?
A:

**MODEL OUTPUT:** The majestic mountains rise up in the distance, their snow-capped peaks glistening in the sunlight. The rushing river flows through the valley, carving its way through the rugged terrain. The answer is no.

---