# OpenReview forum: "How Large Language Models Implement Chain-of-Thought?"
_ICLR.cc/2024/Conference — Submitted to ICLR 2024_

### Official Review · Reviewer_Cn8C · 2023-10-24

**Soundness:** 4 excellent
**Presentation:** 3 good
**Contribution:** 4 excellent
**Rating:** 10
**Confidence:** 5

**Summary:**

This paper makes an important contribution to our understanding of LLMs and how they work. Using reference datasets, this paper looks at the behavior of the model under perturbations of the input data to determine paths and attention heads that are used as parts of reasoning processes. The specific perturbations they use focus on constructing counterfactual examples.

**Strengths:**

The biggest strength of this paper is the question they are asking. While many authors are focused on improving model performance on reasoning tasks, this paper focuses on understanding the internals of the model and advancing the science of the models themselves. The paper was also very strong in its data processing and creation, which was a novel and original reuse of existing data

- Originality: There are two particularly original aspects to this paper. First, the use of reasoning datasets to provide counterfactual examples for models is an original and useful idea. Second

- Quality: Tests were well thought out and explained, datasets were relevant and well chosen. The use of ablation/knockout methods to really focus and prove claims about model performance was particularly nice.

- Clarity: By addressing a hard, technical topic, this paper did not set itself up for success on clarity, however the paper is well written with no major flaws in style or content.

- Significance: As previously stated, the significance of this paper is that it is advancing the science of how LLMs work, rather than improve their performance while punting on the basic understanding of how they work.

**Weaknesses:**

There are two basic weaknesses of this paper. First is clarity, which, as mentioned above, this is an area that it is difficult to be clear in because of the technical nature of the content.  Second, is the generality of the claims they make.

Regarding the generality of the claims, the weakness of this comes from using limited data and reference models. This paper makes claims about LLMs at large, based on two example LLMs. I would like to know why these two models are representative for LLMs at large and why results from these two models are expected to generalize. Even better would be some claims about what classes of models these results are expected to apply to.

**Questions:**

Suggestions:

- Please fix the citation on page 3 for HuggingFace.

- I would also like to see a specific discussion section that pulls together all the results into a summary of what you learned. Right now, that content is spread across a lot of the experimental section, so I'd suggest consolidating it under its own section and highlighting the valuable lessons learned.

---

> ### Author Response · Authors · 2023-11-17
> **Response to Reviewer Cn8C**
>
> We would like to thank the reviewer for taking the time to review our work. We appreciate that you find our paper **ask a good question**.  According to your valuable comments, we provide detailed feedback. Please find the special responses to your comments below.
>
> **Q1**: Clarity issue
> > There are two basic weaknesses of this paper. First is clarity, which, as mentioned above, this is an area that it is difficult to be clear in because of the technical nature of the content.
>
> ***Ans for Q1):***
> Thanks for pointing out the clarity issue.
> - Following your kind suggestion, we have made adjustments to the content of the **Method** section in the main text (highlighted in blue).
> - We have redrawn the framework diagram (Figure 1) of our method and included it in the main text. We would appreciate it if you could find some time to review our revised version.
>
> **Q2**: Generality issue
> > Second, is the generality of the claims they make. Regarding the generality of the claims, the weakness of this comes from using limited data and reference models. This paper makes claims about LLMs at large, based on two example LLMs. I would like to know why these two models are representative for LLMs at large and why results from these two models are expected to generalize. Even better would be some claims about what classes of models these results are expected to apply to.
>
> ***Ans for Q2):***
> Thanks for your constructive comments.
> - The reason we chose these two models (LLaMA2-7B and Qwen-7B) has two folds:
>   - i) The distinguishing feature of large language models compared to earlier language models is their emergent ability, which starts to emerge with 7B parameters.
>   - ii) The model architectures of large language models are predominantly transformers and text data. The main difference between different large language models lies in their training data. LLaMA2-7B [1] is primarily trained on English corpora, while Qwen-7B [1] is trained on both Chinese and English corpora. Both models [1,2] have shown good performance on NLP tasks. Therefore, we chose LLaMA2-7B as a representative of large English language models and Qwen-7B as a representative of large Chinese-English language models.
>
> > ***Reference***
> >
> > [1] Llama 2: Open foundation and fine-tuned chat models. In Arxiv 2023. https://arxiv.org/abs/2307.09288
> >
> > [2] Qwen technical report. https://qianwen-res.oss-cn-beijing.aliyuncs.com/QWEN_TECHNICAL_REPORT.pdf
>
> **Q3**: Typo issue
> > Please fix the citation on page 3 for HuggingFace.
>
> ***Ans for Q3):***
> Thanks for your suggestion and careful review. We have fixed this typo problem.
>
> **Q4**: More discussion
> > I would also like to see a specific discussion section that pulls together all the results into a summary of what you learned. Right now, that content is spread across a lot of the experimental section, so I'd suggest consolidating it under its own section and highlighting the valuable lessons learned.
>
> ***Ans for Q4):***
> Thanks for your careful review and constructive suggestions. In Section 5, we summarized the valuable lessons learned from experiments as follows:
>
> *Moreover, through additional analysis, we discover interesting patterns from the two of the identified key heads specifically. Additionally, through extensive ablation studies on various CE templates and metrics to assess the importance of the patched heads, we find that in-context learning is an effective method to construct REs/CEs for path patching, which can control the behavior of LLMs. The control over LLMs can be strengthened by increasing the few-shot examples in the prompt. Focusing on the probability of the tokens related to a specific task/behavior is a useful method to deal with the sparse causal effect problem.*

---

> ### Author Response · Authors · 2023-11-19
> **Welcome for more discussions (Cn8C)**
>
> Dear reviewer #Cn8C,
>
> Thanks for your valuable time in reviewing and constructive comments, according to which we have tried our best to answer the questions and carefully revise the paper. Here is a **summary of our response** for your convenience:
>
> - (1) **Clarity issue**: Thanks for the kind suggestion, we have made adjustments to the **Method** section and redrawn the framework diagram (**Figure 1**).
> - (2) **Generality issue**: The reason we chose LLaMA2-7B and Qwen-7B has two folds, i) The distinguishing feature, emergent ability, starts to emerge with 7B parameters. ii) The main difference between different LLMs is the training data. LLaMA2-7B is primarily trained on English corpora, while Qwen-7B is trained on both Chinese and English corpora. Both models have shown good performance on NLP tasks. Therefore, we chose LLaMA2-7B as a representative of large English language models and Qwen-7B as a representative of large Chinese-English language models.
> - (3) **Typo issue**: Thanks for your suggestion and careful review. We have fixed this typo problem.
> - (4) **More discussion**: Thanks for your careful review and constructive suggestions. In Section 5 of our revised paper, we summarized the valuable lessons learned from experiments.
>
> We humbly hope our response has addressed your concerns. If you have any additional concerns or comments that we may have missed in our responses, we would be most grateful for any further feedback from you to help us further enhance our work.
>
>
> Best regards
>
> Authors of #4909

---

> ### Author Response · Authors · 2023-11-21
> **Window for discussion and revision is closing**
>
> Dear Reviewer #Cn8C,
>
> Thanks a lot for your time in reviewing and insightful comments, according to which we have carefully revised the paper to answer the questions. We sincerely understand you’re busy. But since the discussion due is approaching, would you mind checking the response and revision to confirm where you have any further questions?
>
> We are looking forward to your reply and are happy to answer your further questions.
>
> Best regards
>
> Authors of #4909

---

> ### Author Response · Authors · 2023-11-22
> **Window for discussion is closing in 21 hours**
>
> Dear Reviewer Cn8C,
>
> Thanks very much for your great efforts in reviewing and valuable comments. The author's discussion will end in the last 21 hours. At this final moment, we sincerely appreciate it if you could check our responses including **Response to Reviewer Cn8C**. And our response to other Reviewers may also provide you with more information.
>
> If you have any further concerns, we will instantly respond to you at this final moment. Your support for a novel discovery in its earlier stage is very important and may inspire more new findings in LLMs interpretability.
>
> Best regards and thanks,
>
> Authors of #4909

---

### Official Review · Reviewer_6hxZ · 2023-10-31

**Soundness:** 3 good
**Presentation:** 2 fair
**Contribution:** 2 fair
**Rating:** 6
**Confidence:** 3

**Summary:**

The paper investigates how chain-of-thought (CoT) prompting enhances reasoning capabilities in large language models (LLMs). The key findings are:
Adjusting the few-shot examples in a CoT prompt is an effective way to generate paired inputs that elicit or suppress reasoning behaviour for analysis. Only a small fraction of attention heads, concentrated in middle and upper layers, are critical for reasoning tasks. Ablating them significantly harms performance. The authors show that some heads focus on the final answer while others attend to intermediate reasoning steps, corresponding to the two stages of CoT.

**Strengths:**

This paper:
- Provides novel insights into CoT reasoning through attention-head analysis.
- Links model components to reasoning subtasks.

**Weaknesses:**

This paper:
- Focuses only on textual reasoning tasks, not more general capabilities.
- Limited to analyzing attention heads, does not cover other components like MLPs.
- Does not modify training to directly improve reasoning abilities.

**Questions:**

Do you think these findings would transfer to more open-ended generative tasks beyond QA?

Did you consider any changes to the model architecture or training process to improve reasoning?

Could your analysis approach help detect if a model is just memorizing vs. logically reasoning?

---

> ### Author Response · Authors · 2023-11-17
> **Response to Reviewer 6hxZ [part 1/2]**
>
> We would like to thank the reviewer for taking the time to review our work. We appreciate that you find our paper provides novel insights into LLMs' CoT reasoning. According to your valuable comments, we provide detailed feedback. Please find the special responses to your comments below.
>
> **Q1**: General capabilities issues:
> > "This paper Focuses only on textual reasoning tasks, not more general capabilities."
>
> ***Ans for Q1):***
> We agree with your perspective that when exploring reasoning ability, it is important to conduct some general investigations.
> - In our study, we followed the work of [1,2,3] and explored the interpretability of the CoT reasoning ability of LLMs using the path patching method they proposed. Our work differs from theirs in that we explore the interpretability of more complex textual reasoning capabilities on a larger language model.
> - As for reasoning abilities in other modalities, that will be part of our future work. We hope that our exploration of the textual modality can inspire research on the interpretability of reasoning abilities in other modalities.
>
> **Q2**:  Not cover MLPs:
> > "This paper limited to analyzing attention heads, does not cover other components like MLPs."
>
> ***Ans for Q2):***
> Thanks for your constructive comments.
> - In this work, we focus on the attention heads because heads can better represent semantic information compared to neurons in MLP layers, consistent with previous work [1].
> - Inspired by your insightful comments, we would like to note that heads serve as input units for MLP. If we can categorize identified heads under different MLPs, we would indirectly identify important MLPs, which may align with your insightful comments.
> - We sincerely appreciate your insight, and in our future work, we will directly explore MLPs rather than analyzing them indirectly through attention heads.
>
>
>
> **Q3**: Model improvement issues:
> > This paper does not modify training to directly improve reasoning abilities.
>
> ***Ans for Q3):***
> Thanks for your insightful comments.
>
> - We mainly follow previous works on LLM interpretability [1,2,3] to locate key heads/units, paving the way for finetuning LLMs.
> - The direction of using identified heads to improve training is indeed promising. However, this exciting direction remains largely under-explored. In this context, we hope our work can contribute to the community in the mentioned promising direction, i.e., interpret-then-finetune.
> - We really appreciate your insightful suggestion to explore how to fine-tune attention heads to enhance model reasoning ability. We will leave it as our future work.
>
> > ***Reference***
> >
> > [1] Interpretability in the Wild: a Circuit for Indirect Object Identification in GPT-2 small. In ICLR 2023.
> >
> > [2] Localizing Model Behavior With Path Patching. In ArXiv 2023.
> >
> > [3] How does GPT-2 compute greater-than?: Interpreting mathematical abilities in a pre-trained language model. In ArXiv 2023.
>
> **Q4**: Generalization issues:
> > Q: Do you think these findings would transfer to more open-ended generative tasks beyond QA?
>
> ***Ans for Q4):***
> Thanks for the inspiring comments.
>
> - We will explore the interpretability of model performance in open-ended generative tasks in future work.
> - We believe that our work can be transferred to more open-ended generative tasks. A possible approach is to perturb the outputs of attention heads and allow the model to generate content freely. By evaluating the changes in the quality of the generated content, we can assess whether a head is crucial for completing the open-ended generative task.
> - However, it is challenging to i) design dynamic counterfactual examples $x_c$ to perform causal intervention and ii) evaluate the quality changes in generated content. Thanks for the inspiring question, we believe exploring these challenges is exciting.
>
> **Q5**: Model improvement issues:
> > Q: Did you consider any changes to the model architecture or training process to improve reasoning?
>
> ***Ans for Q5):***
> Thanks for pointing out the promising direction.
>
> - We agree with your point that identifying key heads and making modifications is a very promising direction. However, this may involve the field of model editing/modifying, which is a promising direction to explore.
> - We would like to note that we mainly focus on how to interpret the reasoning ability of large language models through head localization. Aligning with your insightful question, interpreting models paves the way for model re-training for finetuning to accurately improve model performance, i.e., on a specific task.

---

> ### Author Response · Authors · 2023-11-17
> **Response to Reviewer 6hxZ [part 2/2]**
>
> **Q6**: memorizing vs. reasoning:
> > Q: Could your analysis approach help detect if a model is just memorizing vs. logically reasoning?
>
> ***Ans for Q6):***
> It is an insightful question! Thanks for your kind guidance aiming to make our work more solid.
> - Model memorization is a classical/traditional field in deep learning while exploring a model's memorization capabilities is inherently challenging. In the context of large models, there are two challenges: i) determining whether the model's behavior is due to memorization, generalization, or reasoning is quite a challenging task, and ii) under the conditions of large models, we can access the model weights but not the training data, making it even more difficult to determine whether the model is memorizing or reasoning without access to the data.
>
> We have added a discussion about the interesting direction in our revised paper (Appendix F).

---

> > ### Comment · Reviewer_6hxZ · 2023-11-21
> > **Respond to authors**
> >
> > I thank the authors for responding to my initial comments. I will maintain my score.

---

> > > ### Author Response · Authors · 2023-11-21
> > > **Thanking the Reviewer**
> > >
> > > We sincerely thank the reviewer for your positive feedback on our rebuttal. Many thanks for the kind effort you put into improving the quality of our paper.

---

> ### Author Response · Authors · 2023-11-19
> **Welcome for more discussions (6hxz)**
>
> Dear reviewer #6hxz,
>
> Thanks for your valuable time in reviewing and constructive comments, according to which we have tried our best to answer the questions and carefully revise the paper. Here is a **summary of our response** for your convenience:
>
> - (1) **Focuses only on textual reasoning tasks**: Thanks for the insightful comments, it is important to conduct some general investigations. our work mainly follows the representative work of LLMs interpretability that uses path patching to locate the key component for a specific behavior in LLMs. Exploring reasoning abilities in other modalities will be our future work.
> - (2) **Not cover MLP**: Thanks for your constructive comments. We focus on the attention heads because heads can better represent semantics, which is consistent with previous work. The importance of MLP can be indirectly categorized using the key heads we identified. We will directly explore MLPs in our future work.
> - (3) **Does not modify training**: We mainly follow previous works of LLM interpretability to locate key heads/units. The direction of using identified heads to improve training is promising but largely under-explored. We will leave it as our future work.
> - (4) **Transfer to more open-ended generative tasks**: Thanks for the inspiring comments. We believe that our work can be transferred to more open-ended generative tasks. But there are two challenges: i) design the dynamic $x_c$. ii) evaluate the generated content. In our future work, we will explore the interpretability of the model on generative tasks.
> - (5) **Change the model architecture or training process**: We agree with your point that identifying key heads and making modifications is a very promising direction, which involves the field of model editing and is a promising direction for future work.
> - (6) **Memorizing vs. logically reasoning**: It is an insightful question! Exploring a model’s memorization capabilities is inherently a challenging task. However, since we can not access the model training data, it is even more difficult to determine whether the model is memorizing or reasoning.
>
> We humbly hope our response has addressed your concerns. If you have any additional concerns or comments that we may have missed in our responses, we would be most grateful for any further feedback from you to help us further enhance our work.
>
> Best regards
>
> Authors of #4909

---

> ### Author Response · Authors · 2023-11-21
> **Window for discussion and revision is closing**
>
> Dear Reviewer #6hxZ,
>
> Thanks a lot for your time in reviewing and insightful comments, according to which we have carefully revised the paper to answer the questions. We sincerely understand you’re busy. But since the discussion due is approaching, would you mind checking the response and revision to confirm where you have any further questions?
>
> We are looking forward to your reply and are happy to answer your further questions.
>
> Best regards
>
> Authors of #4909

---

### Official Review · Reviewer_bDfK · 2023-10-31

**Soundness:** 2 fair
**Presentation:** 3 good
**Contribution:** 2 fair
**Rating:** 3
**Confidence:** 4

**Summary:**

This paper aims to explain the CoT reasoning ability of LLMs by identifying "important" attention heads that "contribute the most" to the predictions. Specifically, the paper first constructs counterfactual samples for every referential few-shot CoT sample by replacing the CoT texts in the few-shot examples with random texts generated by ChatGPT. The paper then adopts a method developed in prior work to assign an importance score for every attention head in the LLM.

The authors discover that only a small fraction of the attention heads are important to the CoT task. They also discover that attention heads have different roles: some are responsible for verifying the answer and some are used to synthesize the step-by-step behavior of CoT.

**Strengths:**

Understanding the behavior of LLMs is an important topic that could lead to more robust and trustworthy deep-learning models. This paper focuses on demystifying the chain of thought behavior, which is a practically useful and widely studied phenomenon of LLMs. The paper focuses mainly on identifying important attention heads, which could lead to a better understanding of the attention mechanism employed by Transformer models. Interesting observations regarding the attention patterns and different roles of every attention head have been made.

**Weaknesses:**

My primary concern is that the methodology used in the paper is not tailored to understanding CoT behaviors. Specifically, the method used to identify "important" attention heads is adopted from prior work (Wang et al. (2023) cited in the paper). On the method side, the only task-specific design is how to construct the counterfactual sample $x_c$ given a reference sample $x_r$, which is done by replacing the CoT part in the few-shot example prompt with some randomly generated text (by ChatGPT). It would need more justification why the important attention heads identified by $x_c$ generated in this way contribute to the CoT behavior since (i) it is possible that a (simple) adversarial change in the prompt could significantly decrease the accuracy; (ii) in $x_c$, since the CoT reasoning demonstrations are removed, the LLM would by default not using CoT, which explains the drop in the accuracy; (iii) it would be nice to design the counterfactual example in some other ways, e.g., add incorrect (but still relevant) CoT demonstrations.

Additionally, the score/importance of every attention head is scored by the accuracy drop when substituting its output with the corresponding activations generated by the counterfactual examples. Since $x_r$ and $x_c$ could differ significantly, the paper fails to justify whether replacing the activations directly will have some deteriorating effects on the LLM, since it completely "block" the information flow. This makes it harder to justify the conclusions made in the paper.

**Questions:**

The proposed method does not seem to have specific designs for understanding CoT.

Will directly replacing the activations of certain attention heads have deteriorating effects on the LLM?

Please refer to the weakness section for more information.

---

> ### Author Response · Authors · 2023-11-17
> **Response to Reviewer bDfK [part 1/3]**
>
> We would like to thank the reviewer for taking the time to review our work. We appreciate that you find our paper practically **useful and interesting**. According to your valuable comments, we provide detailed feedback. Please find the special responses to your comments below.
>
>
> **Q1**: Methodology issues:
> > "My primary concern is that the methodology used in the paper is not tailored to understanding CoT behaviors. Specifically, the method used to identify "important" attention heads is adopted from prior work (Wang et al. (2023) cited in the paper). On the method side, the only task-specific design is how to construct the counterfactual sample $x_c$ given a reference sample $x_r$, which is done by replacing the CoT part in the few-shot example prompt with some randomly generated text (by ChatGPT).
> >
> > It would need more justification why the important attention heads identified by generated in this way contribute to the CoT behavior since (i) it is possible that a (simple) adversarial change in the prompt could significantly decrease the accuracy; (ii) in $x_c$, since the CoT reasoning demonstrations are removed, the LLM would by default not using CoT, which explains the drop in the accuracy; (iii) it would be nice to design the counterfactual example in some other ways, e.g., add incorrect (but still relevant) CoT demonstrations."
>
> ***Ans for Q1.1):***
> Thanks for highlighting the potentially confusing issue. We apologize for the misunderstanding, which may result from the inadequate explanation of **path patching**. In response to your comments, we have added corresponding explanations to our work.
>
> - Path patching is a method used to study causal relationships and identify key modules in language models through causal interventions [1,2,3]. The causal interventions are implemented through the design of reference data and counterfactual data $x_r, x_c$, where the key distinction lies in whether they can elicit the model's specific abilities/behaviors. Our focus is on the CoT reasoning ability of large language models (LLMs). In this context, the difference between $x_r, x_c$ lies in whether they can trigger the model's reasoning ability. We believe it aligns with your comments that our $x_r$ triggers the model's CoT ability, and $x_c$ does not. Designing appropriate pairs is challenging and promising for interpreting LLMs.
> - According to path patching, the causal relationship between key heads and the model's CoT ability can be investigated by designing appropriate $x_r, x_c$. The importance of data construction is explicitly stated in path patching [1,2,3], and this crucial aspect is also reflected in our study. Poorly designed $x_c$ may inadvertently trigger the model's reasoning ability, making it difficult to locate the model's CoT reasoning ability, as discussed in our ablation study in Appendix E.
> - The construction of $x_c$ is based on our proposed method that leverages in-context learning for interpretation, which effectively controls the model's reasoning ability while not affecting its other capabilities. As shown in Table 2, when we replace the reasoning process in the few-shot example with irrelevant sentences, the model no longer outputs the reasoning process but still produces the answer.
>
> > ***Reference***
> >
> > [1] Interpretability in the Wild: a Circuit for Indirect Object Identification in GPT-2 small. In ICLR 2023.
> >
> > [2] Localizing Model Behavior With Path Patching. In ArXiv 2023.
> >
> > [3] How does GPT-2 compute greater-than?: Interpreting mathematical abilities in a pre-trained language model. In ArXiv 2023.

---

> > ### Author Response · Authors · 2023-11-17
> > **Response to Reviewer bDfK [part 2/3]**
> >
> > ***Ans for Q1.2):***
> > - i) The impact of adversarial examples on results differs from the effects of causal interventions. Adversarial examples investigate the robustness of the model, whereas our approach based on causal interventions examines the causal relationship between model components and specific outputs [2]. We would like to note that exploring adversarial robustness is also a promising direction, while we mainly focus on interpreting LLMs in this work. Therefore, we mainly focus on interpreting LLMs through a causal-effect view.
> >
> > - ii) We agree with your observation that $x_c$ does not trigger the ability, while $x_r$ does. This is consistent with the motivation to construct these pairs for locating the model's capabilities [1]. Aligning with your insights, our experiments employed $x_c$ that causes limited performance degeneration, i.e., $x_c$ causes a limited drop in the accuracy. Specifically, the accuracy on $x_r$ is $59.5\%$, while the accuracy on $x_c$ is $57.25\%$ (test on 1200 cases of CSQA using LLaMA2-7B).
> >
> > - iii) We appreciate your valuable suggestion. Accordingly, we conduct experiments using the mentioned approach. In our experiments, this approach fails to exhibit sparsity in identifying the relevant heads. Moreover, the knockout of these heads fails to cause a significant impact on the model's CoT reasoning ability, consistent with the results produced by the random approach. We believe this is a valuable exploration, and thus we have provided additional results in **Section 4.4** and **Appendix E**. These results include multiple sets of different methods for constructing $x_r, x_c$.
> >
> > > ***Reference***
> > >
> > > [1] Interpretability in the Wild: a Circuit for Indirect Object Identification in GPT-2 small. In ICLR 2023.
> > >
> > > [2] Localizing Model Behavior With Path Patching. In ArXiv 2023.
> > >
> > > [3] How does GPT-2 compute greater-than?: Interpreting mathematical abilities in a pre-trained language model. In ArXiv 2023.
> >
> > **Q2**: Worry about the conclusion:
> > > "Additionally, the score/importance of every attention head is scored by the accuracy drop when substituting its output with the corresponding activations generated by the counterfactual examples. Since Xr and Xc could differ significantly, the paper fails to justify whether replacing the activations directly will have some deteriorating effects on the LLMs, since it completely "block" the information flow. This makes it harder to justify the conclusions made in the paper."
> >
> > ***Ans for Q2):***
> >
> > We apologize for the misunderstanding, which may results from our insufficient explanation. To address this issue and enhance clarity, we have added the following explanations.
> >
> >
> > - We have provided a further explanation in Section 3.1 in the revised paper (highlighted in blue). Specifically, the key modifications in Section 3.1 are listed below:
> >
> > *Path patching is a method for localizing the key components of LLMs for accomplishing specific tasks or behaviors. The term "behavior" is defined within the context of input-output pairs on a specific dataset, thereby effectively reflecting the model's specific capabilities. To assess the significance of a model component for a particular behavior, a causal intervention is implemented to interfere with the behavior. Assuming that the component holds importance, the causal effect on the model's specific ability should be notable. The causal intervention applied to the model component is achieved through the creation of appropriate reference examples (REs) and counterfactual examples (CEs), which solely vary in their capacity to trigger the model's specific behavior.*
> >
> > - Additionally, we have included a more comprehensible method framework diagram (Figure 1) to explain our methodology.
> >
> > - Regarding the operation of replacing activation, we clarify that it involves replacing the output of a specific head with the output obtained when the input is $x_c$. This operation does not block the information flow but rather replaces key information (while keeping the other information unchanged), which is verified on other tasks like "Indirect Object Identification" [1], and "greater than" [2]. Moreover, the experiment in **Section 4.2** shows that replacing the outputs of random attention heads almost has no effect on the models' CoT reasoning ability, indicating the importance of the identified attention heads.
> >
> > > ***Reference***
> > >
> > > [1] Interpretability in the Wild: a Circuit for Indirect Object Identification in GPT-2 small. In ICLR 2023.
> > >
> > > [2] How does GPT-2 compute greater-than?: Interpreting mathematical abilities in a pre-trained language model. In ArXiv 2023.

---

> ### Author Response · Authors · 2023-11-17
> **Response to Reviewer bDfK [part 3/3]**
>
> **Q3**: Method issue:
> > "The proposed method does not seem to have specific designs for understanding CoT."
>
> ***Ans for Q3):***
> We apologize for the misunderstanding. Accordingly, we would like to highlight the novelty and contribution as follows.
>
> - We have added more explanations (in Section 3.1) to highlight the challenges and two key approaches to address the challenges, aiming to avoid similar issues. Specifically, we propose to leverage in-context learning to construct $x_r, x_c$ (in Section 3.2), and propose a novel metric to evaluate the causal effect, i.e., the proposed Word-of-Interest (WoI) Norm (in Section 3.3). The challenges and our proposed methods in Section 3.1 are shown below, and we believe these changes would make our paper more clear.
> <font color=blue style="font-family: 'Times New Roman';">In this work, we aim to localize the key attention head within the model responsible for CoT reasoning. However, we are faced with two primary challenges: i) The complexity arises from the diverse range of abilities that LLMs possess to achieve CoT reasoning, including numerical computation, knowledge retrieval, and logical reasoning, making it challenging to design reference examples (REs) that are accurately paired with counterfactual examples (CEs) differing only in their ability to trigger the CoT reasoning behavior of LLMs. ii) Additionally, the extensive potential word in the LLM's output results in a sparsity of causal effects, as the words directly related to CoT reasoning are significantly limited in number. To overcome these challenges, we propose an innovative approach for constructing paired REs and CEs, accompanied by a word-of-interest (WoI) normalization method aimed at addressing the issue of sparse causal effects.</font>
>
> - We would like to highlight that i) it is challenging to interpret the CoT reasoning ability, accordingly, we propose to leverage path patching to address the challenge; ii) it is challenging to directly employ path patching for the task, because it is unclear how to construct $x_r, x_c$ to realize path patching, accordingly, we propose an in-context learning approach and a novel approach to realize it; iii) it is challenging to locate key heads even using path patching, accordingly, we propose a novel normalization approach to address the challenge. We believe these explorations are specific designs for understanding CoT.
>
> - Inspired by your valuable comments, we have conducted numerous experiments in data construction and found that utilizing in-context learning is an effective method. Please refer to Section 4.4 for more details about our other attempts to design $x_c$ based on in-context learning. In addition to the in-context learning method, we also attempted to use the zero-shot CoT prompt to design $x_r, x_c$. However, the heads identified through this method are not the key heads for CoT reasoning since knocking out these heads shows little effect on the models' reasoning ability. Detailed results can be found in Appendix E.
>
>
> **Q4**: Worry about the interference with the attention heads:
> > "Will directly replacing the activations of certain attention heads have deteriorating effects on the LLM?"
>
> ***Ans for Q4):***
> Thank you for your valuable question.
> - To validate your viewpoint, we have performed random perturbations on the heads, as shown in **Figure 3**. The random perturbations on random-selected heads have almost no impact on the CoT reasoning capability of the model while perturbing the key heads significantly affects the model's performance. Please refer to **Section 4.2 Validation of key heads** for more detailed information.
> - Inspired by your question, we plan to include the results of model performance on common NLP evaluation tasks (such as MMLU [1]) when knocking out the key heads. We will add it to our work when we complete the experiments.
>
> > ***Reference***
> >
> > [1] Measuring Massive Multitask Language Understanding. In ICLR 2021.

---

> ### Author Response · Authors · 2023-11-19
> **Welcome for more discussions (bDfK)**
>
> Dear reviewer #bDfK,
>
> Thanks for your valuable time in reviewing and constructive comments, according to which we have tried our best to answer the questions and carefully revise the paper. Here is a **summary of our response** for your convenience:
>
> - (1) **Methodology issues**: Thanks for highlighting the potentially confusing issue. we have explained our **methodology** in ***Answer for Q1.1*** and listed some representative papers of LLM interpretability. We have explained the relationship between adversarial change and causal intervention in ***Answer for Q1.2*** for better understanding. Furthermore, according to your suggestion about $x_c$ design, extensive experiments about $x_r, x_c$ construction are discussed in **Section 4.4** and **Appendix E**.
> - (2) **Deteriorating effects on LLMs**: Following your constructive advice, a further explanation about path patching and a more comprehensible method framework diagram have been supplemented in the revised paper. In **Section 4.2**, we have verified that replacing the activations directly will not have deteriorating effects on the LLMs.
> - (3) **No specific designs for understanding CoT**: Apologies for the misunderstanding, more explanations about the challenges and our innovative methods are highlighted and added to **Section 3.1**. Inspired by your valuable comments, numerous experiments in data construction have been further conducted and we find that utilizing in-context learning is an effective method.
> - (4) **Worry about the interference with the attention heads**: Thank you for your valuable question. We have performed random perturbations on the heads. The random perturbations have almost no impact on the model on CoT reasoning while perturbing the key heads significantly affects the model’s performance. (See **Section 4.2  Validation of key heads)
>
> We humbly hope our response has addressed your concerns. If you have any additional concerns or comments that we may have missed in our responses, we would be most grateful for any further feedback from you to help us further enhance our work.
>
>
> Best regards
>
> Authors of #4909

---

> ### Author Response · Authors · 2023-11-21
> **Window for discussion and revision is closing**
>
> Dear Reviewer #bDfK,
>
> Thanks a lot for your time in reviewing and insightful comments, according to which we have carefully revised the paper to answer the questions. We sincerely understand you’re busy. But since the discussion due is approaching, would you mind checking the response and revision to confirm where you have any further questions?
>
> We are looking forward to your reply and are happy to answer your further questions.
>
> Best regards
>
> Authors of #4909

---

> ### Author Response · Authors · 2023-11-22
> **Window for discussion is closing in last 21 hours**
>
> Dear Reviewer bDfK,
>
> Thanks very much for your great efforts in reviewing and valuable comments. The author's discussion will end in the last 21 hours. At this final moment, we sincerely appreciate it if you could check our responses including **Response to Reviewer bDfK [part 1/3]**, **Response to Reviewer bDfK [part 2/3]**, and **Response to Reviewer bDfK [part 3/3]**. And our response to other Reviewers may also provide you with more information.
>
> If you have any further concerns, we will instantly respond to you at this final moment. And we sincerely appreciate that you could consider improving your score if there is no further concern. Your support for a novel discovery in its earlier stage is very important and may inspire more new findings in LLMs interpretability.
>
> Best regards and thanks,
>
> Authors of #4909

---

> ### Author Response · Authors · 2023-11-23
> **Window for responsing and draft updating is closing**
>
> Dear Reviewer #bDfK,
>
> Thanks a lot for your time in reviewing and reading our response and the revision. Thanks very much for your valuable comments. We sincerely understand you’re busy. But as the window for responding and paper revision is closing, would you mind checking our response (a brief summary, and details) and confirming whether you have any further questions? We look forward to answering more questions from you.
>
> Best regards and thanks,
>
> Authors of #4909

---

### Official Review · Reviewer_wEdy · 2023-10-31

**Soundness:** 3 good
**Presentation:** 2 fair
**Contribution:** 3 good
**Rating:** 6
**Confidence:** 4

**Summary:**

The reasoning processes of the LLaMA2-7B and Qwen-7B models are interpreted using the path patching method (initially introduced in [1], which is an interoperability method rooted in causal intervention), in the context of using few-shots prompts. The evaluation is grounded on three benchmarks: StrategyQA, AQuA, and CSQA (which have all been introduced in previous publications). They find that only a small number of attention heads are responsible for reasoning, and they also find that they are located in specific locations in the model's architecture.

This represents a solid effort to interpret large language models using the path patching method to date and is the first paper where the chain-of-thoughts method is interpreted using the path patching method.

[1] https://openreview.net/pdf?id=NpsVSN6o4ul

**Strengths:**

Carrying out this piece of research requires handling several different technical aspects (carefully constructing the datasets to allow counterfactual evaluation, applying the path patching method, and evaluating the effects of knocking out different attention heads), which the authors seem to largely have done well, including paying attention to subtle issues like the choice of right metric (section 4.4.2).

Their findings highlight interesting behaviors that the attention heads display, which is summarized in section 4.3, for example: "_Analogically,
the function of Head 18.30 and 13.16 corresponds to the two stages of the chain-of-thought (CoT) process: firstly think step-by-step to get intermediate thoughts, then answer the question based on these thoughts._"

I think such research will become more widely spread in the future in order to understand the reasoning processes of attention-based models better, and as such, it is very timely work.

**Weaknesses:**

- the presentation is at times unclear (see the questions section). I found it quite hard to read and had to spend some time understanding their methodology
- the literature review section could be more comprehensive: a number of other articles on interpretability rooted in causal interventions exist on models of similar sizes, such as [2,3], the latter also using a 7B model but not being cited. I would recommend the authors contrast the existing approaches in the "_Related Work_" section or an appendix to that section, such as from [3] -but there are also other articles- with their own and comment upon similarities and differences so that the reader is well-informed of how their methods compete with existing ones.
- the improvements are nice but somewhat incremental since a single in-context learning technique is analyzed on just two models.
- their methodology might also be improved by the use of diagrams to show in a single glance all the relevant information
- it's somewhat strange that Appendix B ("_Reference Data Examples_") is empty; why include it in that case?


[2] https://arxiv.org/pdf/2305.00586.pdf
[3] https://arxiv.org/pdf/2305.08809.pdf

**Questions:**

Tables 1, 2 / 4, 5 are unclear and suffer from a number of issues:
- Table 1, 2: A reader might at first be confused whether what is shown is the complete few-shot that is supplied to the model that is to be tested (e.g., LLaMA2) to facilitate in-context learning - or if what follows after the "A" is the answer of LLaMA2/Qwen?
That the former is correct transpires indirectly only from the text: "_The outputs of LLaMA-7B and Qwen-7B on the counterfactual data are shown in Table 4 and 5, respectively._" I would recommend adding at least a caption here, explaining more clearly what can be seen, without having to look in the text for the meaning of the table.
- In Table 1,2, the last question is "_Can Reiki be stored in a bottle?_", which led me to believe that this is the question the model should have answered (once with the correct in-context learning text and once with the modified/colored text as indicated from table 2).
But a look in the appendix at Table 4 suddenly reveals a different last question, "_Would a rabbi worship martyrs Ranavalona I killed?_" (all else being the same), which I found confusing. Can the authors explain this?
- minor formatting: "Q" and "A" from Table 1,2 are set in bold but not in Table 4,5, which does not aid readability.
- since these tables seem an essential part of the paper, as the capture the methodology, it is tiring to go back and forth between Tables 1,2 and the Appendix; I would propose to move all Tables to the main body to aid readability.

In the section "Counterfactual data generation" you say: "_x_c is generated by partially editing x_r with the purpose of altering the model’s reasoning effects. In order to suppress the reasoning capability of the model while maintaining equal lengths for counterfactual data and   reference data, we replaced the evidence in x_r with irrelevant sentences describing natural scenery, which were randomly generated
using ChatGPT-3.5._"
No mention of a specific dataset is being made. Shouldn't the change that you made account for the specific structure of the dataset? I am guessing if some dataset actually deals with natural scenery, then inserting random natural scenery descriptions might not achieve the desired counterfactual effect. Some statement should be made here that what is randomly included matches the dataset.

**Details Of Ethics Concerns:**

(not applicable)

---

> ### Author Response · Authors · 2023-11-17
> **Response to Reviewer wEdy [part 1/3]**
>
> We would like to thank the reviewer for taking the time to review our work. We appreciate that you find our paper is **a solid effort** to interpret LLMs and our method is carefully designed. According to your valuable comments, we provide detailed feedback. Please find the special responses to your comments below.
>
>
> **Q1**: Presentation issues:
> > "The presentation is at times unclear (see the questions section). I found it quite hard to read and had to spend some time understanding their methodology."
>
> ***Ans for Q1):***
> Inspired by your valuable comments, we have made the following improvements/explanations:
>
> **a)** We have adjusted the misleading Table 1,2/4,5, and the specific modifications will be explained in detail in ***Ans for Q6)***.
>
> **b)** We have redrawn the methodology framework diagram (Figure 1) and included it in the main text instead of the appendix.
>
> **c)** We have made adjustments to the content of the **Method** section in the main text and highlighted the specific changes in blue. We would appreciate it if you could find some time to review our revised version paper, which incorporates these changes.
>
> Specifically, in order to enhance the clarity of the methodology, we have reorganized the **Method** section. We now start by introducing the core framework of the path patching method and have added an introduction to path patching in Sec.3.1:
>
>
> *Path patching is a method for localizing the key components of LLMs for accomplishing specific tasks or behaviors. The term "behavior" is defined within the context of input-output pairs on a specific dataset, thereby effectively reflecting the model's specific capabilities. To assess the significance of a model component for a particular behavior, a causal intervention is implemented to interfere with the behavior. Assuming that the component holds importance, the causal effect on the model's specific ability should be notable. The causal intervention applied to the model component is achieved through the creation of appropriate reference examples (REs) and counterfactual examples (CEs), which solely vary in their capacity to trigger the model's specific behavior.*
>
> and supplement the challenges of interpreting LLMs' CoT reasoning ability using path patching in Sec.3.1:
>
> *In this study, we aim to localize the key attention head within the model responsible for CoT reasoning. However, we are faced with two primary challenges: i) The complexity arises from the diverse range of abilities that LLMs possess to achieve CoT reasoning, including numerical computation, knowledge retrieval, and logical reasoning, making it challenging to design reference examples (REs) that are accurately paired with counterfactual examples (CEs) differing only in their ability to trigger the CoT reasoning behavior of LLMs. ii) Additionally, the extensive potential word in the LLMs' output results in a sparsity of causal effects, as the words directly related to CoT reasoning are significantly limited in number. To overcome these challenges, we propose an innovative approach for constructing paired REs and CEs, accompanied by a word-of-interest (WoI) normalization method aimed at addressing the issue of sparse causal effects.*
>
> This sets the stage for the subsequent discussion of our method's innovations, namely the $x_r, x_c$ construction, and causal effect evaluation metric design.

---

> > ### Author Response · Authors · 2023-11-17
> > **Response to Reviewer wEdy [part 2/3]**
> >
> > **Q2**: Literature review issues:
> > > "the literature review section could be more comprehensive: a number of other articles on interpretability rooted in causal interventions exist on models of similar sizes, such as [2,3], the latter also using a 7B model but not being cited. I would recommend the authors contrast the existing approaches in the "Related Work" section or an appendix to that section, such as from [3] -but there are also other articles- with their own and comment upon similarities and differences so that the reader is well-informed of how their methods compete with existing ones.
> > the improvements are nice but somewhat incremental since a single in-context learning technique is analyzed on just two models.
> >
> >
> > ***Ans for Q2):***
> > Thank you for providing these outstanding related works. Based on your suggestions, we have incorporated citations to these papers in our paper and discussed them in Sec.2 Background as follows:
> >
> > *[1] takes the first step toward understanding the working mechanism of the 7B-sized large language model. Method Distributed Alignment Search (DAS) [2] based on causal abstraction is applied to align the language model with a hypothesized causal model.*
> >
> > *Due to the complexity of CoT reasoning tasks, it is intricate to design a unified symbolic causal model for multi-step reasoning tasks. In this work, employing the path patching method from [3,4], we successfully identify the crucial attention heads in LLMs responsible for CoT reasoning ability.*
> >
> > We have added these contents to Section 2 of our revised paper and modifications are highlighted.
> >
> > > ***Reference***
> > >
> > > [1] Interpretability at Scale: Identifying Causal Mechanisms in Alpaca. In ArXiv 2023.
> > >
> > > [2] Finding Alignments Between Interpretable Causal Variables and Distributed Neural Representations. In NeurIPS 2023.
> > >
> > > [3] How does GPT-2 compute greater-than?: Interpreting mathematical abilities in a pre-trained language model. In ArXiv 2023.
> > >
> > > [4] Interpretability in the Wild: a Circuit for Indirect Object Identification in GPT-2 small. In ICLR 2023.
> >
> > **Q3**: Experiment issues:
> > > "the improvements are nice but somewhat incremental since a single in-context learning technique is analyzed on just two models."
> >
> > ***Ans for Q3):***
> > We really appreciate your feedback. It is important to support our points with comprehensive experiments. In response to your comments, we have supplemented two additional templates (**Section 4.4 template 1&2**) for constructing counterfactual data adopting in-context learning. The key heads identified using these templates are shown in Figure 5b and c, which are similar to the heads identified using the template we originally proposed (Fig. 5a).
> >
> > Additionally, inspired by your valuable suggestion, we have explored using zero-shot CoT to construct $x_r, x_c$ instead of in-context learning (Appendix E). However, we find that the identified heads using this approach are not crucial for the model's reasoning capability, as excluding these heads does not significantly impact the model's reasoning performance (please refer to Appendix E for more results).
> >
> > Furthermore, drawing inspiration from your suggestion, we conducted another interesting experiment. We noticed that the same key head (Head 18.30 for LLaMA2-7B) is identified on the CSQA and AQuA datasets, but not on the StrategyQA dataset (see Figure 2). The difference between these datasets is that $x_r, x_c$ of CSQA and AQuA have multiple-choice question formats, whereas those of StrategyQA have a true or false question format. Therefore, we converted the $x_r, x_c$ on the StrategyQA dataset into the multiple-choice format. (Please refer to **Template 3** in Section 4.4 for specific examples.)
> >
> > As a result, we identified the same key head (Head 18.30) as on CSQA and AQuA datasets (see Fig. 5a, d). Further analysis and details are presented in Section 4.4.
> >
> > **Q4**: Methodology issues:
> > > "their methodology might also be improved by the use of diagrams to show in a single glance all the relevant information."
> >
> > ***Ans for Q4):***
> > Thanks for your suggestion. We have redrawn the method figure and moved it to **Section 3** (**Figure 1**).
> >
> > **Q5**: Typo issues:
> > > "it's somewhat strange that Appendix B ("Reference Data Examples") is empty; why include it in that case?"
> >
> > ***Ans for Q5):***
> > Thanks for your suggestion and careful review. We have fixed this typo problem.

---

> > > ### Author Response · Authors · 2023-11-17
> > > **Response to Reviewer wEdy [part 3/3]**
> > >
> > > **Q6**: Table issues:
> > > > "Tables 1, 2 / 4, 5 are unclear and suffer from a number of issues:
> > > > 1. Table 1, 2: A reader might at first be confused whether what is shown is the complete few-shot that is supplied to the model that is to be tested (e.g., LLaMA2) to facilitate in-context learning - or if what follows after the "A" is the answer of LLaMA2/Qwen? That the former is correct transpires indirectly only from the text: "The outputs of LLaMA-7B and Qwen-7B on the counterfactual data are shown in Table 4 and 5, respectively." I would recommend adding at least a caption here, explaining more clearly what can be seen, without having to look in the text for the meaning of the table.
> > > > 2. In Table 1,2, the last question is "Can Reiki be stored in a bottle?", which led me to believe that this is the question the model should have answered (once with the correct in-context learning text and once with the modified/colored text as indicated from table 2). But a look in the appendix at Table 4 suddenly reveals a different last question, "Would a rabbi worship martyrs Ranavalona I killed?" (all else being the same), which I found confusing. Can the authors explain this?
> > > > 3. minor formatting: "Q" and "A" from Table 1,2 are set in bold but not in Table 4,5, which does not aid readability.
> > > > 4. since these tables seem an essential part of the paper, as the capture the methodology, it is tiring to go back and forth between Tables 1,2 and the Appendix; I would propose to move all Tables to the main body to aid readability."
> > >
> > > ***Ans for Q6):***
> > > Thanks for your careful review. According to your valuable suggestion, we have moved Table 1 and Table 2 from the appendix to Section 3.2. To avoid misleading, each table now includes two parts, one for the model's input sentences and the other for the output sentences. Additionally, we have standardized the questions used for $x_r, x_c$ in Table 1 and Table 2 with the questions from the CSQA dataset. Due to space limitations, we have relocated the examples of $x_r, x_c$ on the StrategyQA and AQuA datasets to Table 3 and Table 4 in Appendix B, and they are labeled in the same format as Table 1 & 2 with model input and output. Concerning the formatting issue with bolding "Q" and "A", we have made the necessary corrections.
> > >
> > > **Q7**: data issues:
> > > > "In the section "Counterfactual data generation" you say: "x_c is generated by partially editing x_r with the purpose of altering the model’s reasoning effects. In order to suppress the reasoning capability of the model while maintaining equal lengths for counterfactual data and reference data, we replaced the evidence in x_r with irrelevant sentences describing natural scenery, which were randomly generated using ChatGPT-3.5." No mention of a specific dataset is being made. Shouldn't the change that you made account for the specific structure of the dataset? I am guessing if some dataset actually deals with natural scenery, then inserting random natural scenery descriptions might not achieve the desired counterfactual effect. Some statement should be made here that what is randomly included matches the dataset."
> > >
> > > ***Ans for Q7):***
> > > Thanks for your valuable advice. A statement about the relationship between the generated irrelevant sentences and three CoT reasoning datasets (StrategyQA, CSQA, and AQuA) has been made in the footnote on page 5. A specific statement is listed below:
> > >
> > > *We ensure that the generated irrelevant sentences are simple declarative statements and do not involve any inference or coherence relationships between them. Furthermore, the reasoning involved in the dataset we use is unrelated to natural landscapes, which mainly focuses on mathematical, commonsense, and historical reasoning. Therefore, the likelihood of the replaced content affecting the reasoning outcomes is minimal.*

---

> ### Author Response · Authors · 2023-11-19
> **Welcome for more discussions (wEdy)**
>
> Dear reviewer #wEdy,
>
> Thanks for your valuable time in reviewing and constructive comments, according to which we have tried our best to answer the questions and carefully revise the paper. Here is a **summary of our response** for your convenience:
>
> - (1) **Unclear presentation**: Following you constructive comments, we have reorganized our **method section** and provided additional explanation about path patching.
> - (2) **Literature issues**: Following your constructive comments, we have discussed related works based on causal abstraction. And we also add these discussions into our revision to enhance our work.
> - (3) **A single in-context learning technique is analyzed**: We agree with your valuable suggestions, we have supplemented the results of experiments that used zero-shot CoT to construct $x_r,x_c$, and experiments that used other in-context learning methods to construct $x_c$. In addition, an interesting experiment is conducted (please see our ***Answer for Q3***).
> - (4) **Improve the methodology and empty Appendix B**:  Thanks for your valuable advice, we have redrawn our figure of method overview and moved the figure to Section 2.
> - (5) **empty Appendix B**: Thanks for your careful review, we have fixed the typo.
> - (6) **Table issues**: For our misleading Table 1,2/3,4, we have adjusted the content of the tables and added the caption to help better understand our data.
> - (7) **No mention of a specific dataset**: According to your valuable device, we have provided more explanations to illustrate the relationship between the generated irrelevant sentences and the CoT reasoning datasets we used.
>
> We humbly hope our response has addressed your concerns. If you have any additional concerns or comments that we may have missed in our responses, we would be most grateful for any further feedback from you to help us further enhance our work.
>
>
> Best regards
>
> Authors of #4909

---

> ### Author Response · Authors · 2023-11-21
> **Window for discussion and revision is closing**
>
> Dear Reviewer #wEdy,
>
> Thanks a lot for your time in reviewing and insightful comments, according to which we have carefully revised the paper to answer the questions. We sincerely understand you’re busy. But since the discussion due is approaching, would you mind checking the response and revision to confirm where you have any further questions?
>
> We are looking forward to your reply and are happy to answer your further questions.
>
> Best regards
>
> Authors of #4909

---

> ### Author Response · Authors · 2023-11-22
> **Window for discussion is closing in last 21 hours**
>
> Dear Reviewer wEdy,
>
> Thanks very much for your great efforts in reviewing and valuable comments. The author's discussion will end in the last 21 hours. At this final moment, we sincerely appreciate it if you could check our responses including **Response to Reviewer wEdy [part 1/3]**, **Response to Reviewer wEdy [part 2/3]**, and **Response to Reviewer wEdy [part 3/3]**. And our response to other Reviewers may also provide you with more information.
>
> If you have any further concerns, we will instantly respond to you at this final moment. And we sincerely appreciate that you could consider improving your score if there is no further concern. Your support for a novel discovery in its earlier stage is very important and may inspire more new findings in LLMs interpretability.
>
> Best regards and thanks,
>
> Authors of #4909

---

> ### Comment · Reviewer_wEdy · 2023-11-23
> **Reviewer answer**
>
> I thank the authors for their thorough effort in clarifying all points and presenting a significantly updated revision of their paper.
>
> Some points: Figure 1 looks much clearer now, but I would advise the authors to use slightly darker colors from green and red, since text such as "Hydrogen is the first element and has an atomic number of one. To square a number, you multiply it by itself. The Spice Girls has five members" is hard to read.
> I appreciated the inclusion of Figure 6 related to the knockout method, which clarifies further their methodology (I think this figure is new; I cannot access the original paper version to compare under "Revisions").
> Also, a number of improvements have been made throughout the paper, and new experiments have been carried out.
>
> All in all, I am satisfied that this is a substantially improved version of the paper. It would now be a solid 7 (from an original 6); unfortunately, ICLR does not allow scores of 7 (only 6 and 8), so I am keeping my score unchanged.

---

> > ### Author Response · Authors · 2023-11-23
> > **Thanks for your swift reply and raising the score**
> >
> > Dear Reviewer #wEdy,
> >
> > Thanks for your swift reply despite such a busy period. We sincerely appreciate that you carefully reviewed our revised paper and you can raise the score. Thanks for your valuable suggestion, we are now adjusting the text color in our methodology figure and we will update the revised paper soon. If you have any further questions or comments, we are very glad to discuss more with you. Your valuable comments have greatly helped us in enhancing our work.
> >
> > Best regards and thanks,
> >
> > Authors of #4909

---

### Meta-Review · Area_Chair_cre4 · 2023-12-11

**Metareview:**

This paper is an attempt at understanding the mechanism behind chain-of-thought (CoT) reasoning. The paper generates reference (RE) and counterfactual (CE) examples via few-shot prompts, and uses path patching to find the attention heads most important in carrying out CoT reasoning. The experiments found that only a small fraction of heads in mid to late layers play a critical role in CoT via two distinct roles: aiding 1) reasoning steps 2) getting to the final answer.

The problem this paper tackles, understanding the mechanisms by which transformer language models do “reasoning”, is important and timely. Most reviewers found this paper well-executed. In addition, the insights this paper presents are novel and interesting.

Constructive engagement during the discussion period has led to improvements to the manuscript, such as clearer explanation of the method section. Positioning with respect to prior work has been improved, as well.

However, there are a few issues with the current version of the draft. There are concerns raised about how counterfactuals are made from references, which is done by replacing the CoT part of a few shot prompt with randomly generated text. Using in-context-learning only with a few templates for creating RE and CEs might be limiting as raised by another review. Authors have conducted additional experiments with zero-shot prompts as alternative approaches to generate REs and CEs. However, the results are not replicated which might cast doubt on the robustness of the findings, and whether they are artifacts of the ICL setup used to curate CEs and REs. Reviewers were concerned about the novelty of the methodology. It is worth noting that building on top of prior methods is rather encouraged, and carrying out a carefully designed experiment has significant value. However, when the bulk of the contributions are about the experimental insights, a more in-depth analysis is required to assure that the insights are not artifacts of experimental design choices.

Other limitations were discussed, but they can be deferred to future work (e.g. generalization to other generative tasks beyond QA or modalities other than text, or using these findings to improve reasoning further, memorization vs actual reasoning).

Overall, the paper can benefit from another round of revision to further ensure the robustness of these findings. This could also be an opportunity to further improve the presentation of this work.

**Justification For Why Not Higher Score:**

- Insights might be artifacts of limited experimental setups of CE and RE curation
- The presentation has led to many confusions. There have been significant improvements over the discussion period, but it can benefit from more clarification.

**Justification For Why Not Lower Score:**

N/A

---

### Decision · Program_Chairs · 2024-01-16

Reject